# RNNs perform task computations by dynamically warping neural representations

**Arthur Pellegrino**
The Gatsby Unit
University College London
a.pellegrino@ucl.ac.uk

**Angus Chadwick**
School of Informatics
University of Edinburgh
angus.chadwick@ed.ac.uk

## Abstract

Analysing how neural networks represent data features in their activations can help interpret how they perform tasks. Hence, a long line of work has focused on mathematically characterising the geometry of such "neural representations." In parallel, machine learning has seen a surge of interest in understanding how dynamical systems perform computations on time-varying input data. Yet, the link between computation-through-dynamics and representational geometry remains poorly understood. Here, we hypothesise that recurrent neural networks (RNNs) perform computations by dynamically warping their representations of task variables. To test this hypothesis, we develop a Riemannian geometric framework that enables the derivation of the manifold topology and geometry of a dynamical system from the manifold of its inputs. By characterising the time-varying geometry of RNNs, we show that dynamic warping is a fundamental feature of their computations.

## 1 Introduction

Understanding the low-dimensional neural representations used by biological and artificial neural networks to solve complex tasks is a key challenge in machine learning.[1,2] For example, low-dimensional activity manifolds have been observed in neural systems performing vision[3,4] or language tasks.[5] More generally, data from the natural world often lie on low-dimensional manifolds,[6] and the computations performed by neural networks trained on these data can be interpreted as transformations of these data manifolds into neural activation manifolds.

However, existing work on the geometry of neural representations has focused on the *static*, time-independent, setting.[7] In contrast, the geometry of the dynamic representations formed by neural networks driven by time-varying inputs remains largely unexplored.

Dynamical system models, described as differential equations, are ubiquitous in science and machine learning. The development and application of methods to infer dynamical systems from data[8–10] or train dynamical systems to solve tasks[11] are now central to computational neuroscience,[12–14] physics,[15] chemistry[16] and genomics.[17,18] In machine learning, diffusion models based on stochastic differential equations achieve state-of-the-art performance for generative modelling of images.[19] In computational neuroscience, recurrent neural networks (RNNs) defined using differential equations are used routinely to investigate neural computations underlying various behavioural tasks.[20]

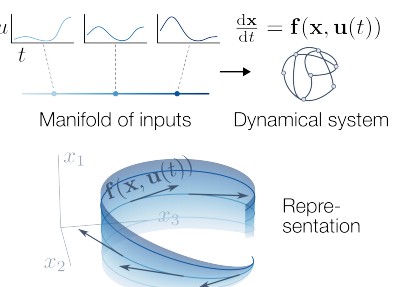

Figure 1: **Dynamical systems receiving inputs on a low-dimensional manifold of functions are constrained to low-dimensional manifold of states.** The Riemannian geometry of this manifold can provide insights into dynamical representations and computations.

39th Conference on Neural Information Processing Systems (NeurIPS 2025).

However, both data-driven and task-optimised nonlinear dynamical systems models are typically viewed as "black boxes", with their behaviour lacking a clear and interpretable explanation.[21–23] Attempts to interpret dynamical systems models typically focus on the topology and geometry of fixed point manifolds, and the locally linearised dynamics about them.[21] This approach has provided insight into the computations used by RNNs to solve a range of tasks.[24] Nevertheless, the study of dynamics near fixed points faces a number of limitations. First, many systems exhibit complex, time-varying activity patterns which do not settle into a fixed point,[25,26] or nonlinear dynamics before settling into a fixed point which are crucial to the computation.[27–29] Second, the characterisation of fixed point manifolds is insufficient to determine the geometry and topology of the full activity manifold, and therefore provides an incomplete understanding of the system.

Thus, two key questions remain. First, what is the relationship between the geometry and topology of the data manifold and that of the dynamical system's activity, and how does this relate to the internal dynamics of the system? Second, in the case where the dynamical system is trained to perform a specific task, as are RNNs, how can these geometric and topological properties be related to the computations performed by the network to solve the task?

Here, we show that Riemannian geometry can be leveraged to understand the neural representations used by dynamical systems to perform time-dependent task computations. Specifically, we prove that under weak constraints the dynamical system's state is restricted to a manifold whose topology is closely related to that of the input data manifold. Since the input manifold is independent of the trained parameters of the model, the geometry of the neural activity manifold reflects the computation it performs on these inputs. To analyse this geometry, we derive the pullback Riemannian metric on the neural manifold of a general class of dynamical systems. This metric naturally describes the local encoding of task variables and is defined via an adjoint differential equation. Using this metric we demonstrate that several classic RNN models from neuroscience solve tasks by dynamically warping their internal representations of input variables. Overall, we offer a novel formal mathematical framework to characterise the geometry of dynamical systems beyond attractor manifolds, which can be used to study RNN representations and computations.

## Contributions

- We derive a relationship between the topology of the manifold of inputs to a dynamical system and the manifold over which the state of the system is constrained. We prove that for inputs on a $m$-dimensional manifold, the state of the system is constrained to an $m + 1$-dimensional manifold.

- We derive the pullback of the metric, from the neural state-space to the input manifold for a general class of dynamical systems. In several examples we show that this metric is a reflection of the computations performed by the dynamical system on its inputs to solve a task.

- In an RNN trained on a contextual decision-making task we show that, over time, the manifold becomes warped so as to compress irrelevant input information. In a sequential working memory task, we show that RNN activity lies on a hyper-torus, whose extrinsic and intrinsic geometry dynamically warps to retrieve different memories at different points in time.

## Related works

**Attractor manifolds and dynamical systems interpretability.** Attractor manifolds of dynamical system models have been widely studied to interpret the underlying computations of RNNs,[30,31] denoising diffusion models[32,33] and neuralODEs[34] in tasks spanning language[35] reinforcement learning[33] and computational neuroscience.[21,36] Here instead, we show that the computations performed by a dynamical system are reflected by the geometry of the manifold over which its state lies. Works beside ours have suggested going beyond the study of fixed points, for example via distillation of nonlinear dynamics to a low-dimensional subspace,[37] or by applying dimensionality reduction methods to the systems' dynamics[23,38] or trajectories.[39,40] In contrast, we show that the low-dimensional geometry of the system can be exactly mathematically derived via an adjoint dynamical system.

**Riemannian neural representations.** The Riemannian geometry of neural activity in deep networks has been extensively studied.[7,41–43] This has helped interpret how neural networks represent features of images,[44,45] RL task variables[46] and text[47] in their neural activity. However, existing frameworks typically assume static inputs. Here, we mathematically characterise the dynamic geometry of neural networks described via differential equations receiving time-varying inputs.

## 2 Riemannian geometry can link representation and computation

**Review of Riemannian geometry.** Before studying the manifolds of dynamical systems, we first consider a *static* deep neural network to provide insight into a simplified setting. Throughout, we work with a function $\varphi : \mathcal{M} \to \mathbb{R}^n$ mapping inputs on a data manifold $\mathcal{M}$ to an ambient space $\mathbb{R}^n$. In the case where $\varphi$ is a deep neural network, this ambient space may be that of the $n$ neurons in one of its hidden layers. In the next section, we generalise this setting to the time-varying case, where $\varphi$ is the solution to a dynamical system, and $\mathbb{R}^n$ its state-space.

Following previous work,[7,41,42] we can derive a metric on $\mathcal{M}$, called the *pullback metric*, which captures the change in geometry generated by the neural network $\varphi$. We denote the tangent space at a point $p \in \mathcal{M}$ as $T_p\mathcal{M}$. If $\varphi$ is differentiable and $\mathrm{d}\varphi : T_p\mathcal{M} \to \mathbb{R}^n$ is injective then any metric defined on $\mathbb{R}^n$ can be "pulled back" to obtain a metric on $\mathcal{M}$. One such metric is the standard dot product: $\langle \cdot, \cdot \rangle : \mathbb{R}^n \times \mathbb{R}^n \to \mathbb{R}$. The metric inherited by $\mathcal{M}$ from $\mathbb{R}^n$ via pullback is defined as:

$$g : T_p\mathcal{M} \times T_p\mathcal{M} \xrightarrow{\mathrm{d}\varphi, \mathrm{d}\varphi} \mathbb{R}^n \times \mathbb{R}^n \xrightarrow{\langle \cdot, \cdot \rangle} \mathbb{R}.$$

In words, the inner product between tangent vectors of $\mathcal{M}$ can be measured by first mapping these vectors to $\mathbb{R}^n$, and then taking their usual Euclidean dot product. If the manifold $\mathcal{M}$ is endowed with local coordinates $(x_1, ..., x_m)$, the tangent space at a point $p \in \mathcal{M}$ has a natural basis $\mathbf{v}_1(p) = \frac{\partial}{\partial x_1}, ..., \mathbf{v}_m(p) = \frac{\partial}{\partial x_m} \in T_p\mathcal{M}$, corresponding to infinitesimal changes on the manifold as a function of infinitesimal changes in any of the local coordinates. In that case, the metric $g$ can be written, at a particular point, as a matrix $G(p) \in \mathbb{R}^{m \times m}$ whose $(i,j)$th entry corresponds to the inner product between $\mathbf{v}_i(p)$ and $\mathbf{v}_j(p)$, that is $G_{ij}(p) = g(\mathbf{v}_i(p), \mathbf{v}_j(p)) = (\mathrm{d}\varphi(p)\mathbf{v}_i(p)) \cdot (\mathrm{d}\varphi(p)\mathbf{v}_j(p))$.

Importantly, the metric $g$ carries all the information about the intrinsic geometry of the manifold $\varphi(\mathcal{M}) \subseteq \mathbb{R}^n$. For example, it captures the local curvature and warping of the geometry of the data manifold induced by the mapping $\varphi$. We next show how the *intrinsic geometry* of the hidden-layer manifold $\varphi(\mathcal{M})$ can provide insights into the *computations* performed by he network.

**Deep neural network representations.** We start by providing intuition in the setting of deep neural networks with no time-dependence. To this end, we built a simple feedforward network trained on a binary classification task. The inputs were drawn from a circular manifold – embedded in a 2D plane – and the network was trained to map angles $\theta \in [0\,\pi)$ to 1 and $\theta \in [\pi, 2\pi)$ to $-1$ (Fig. 2a; S8.1). A simple 3-neuron network is sufficient to solve this task, and visualising the activation the neurons in response to all inputs showed that the network learned a non-trivial representation of the input manifold (Fig. 2b).

Since this is a 1-d manifold, the metric has a single entry describing the magnitude of the change in neural activation given a small change in the angle: $G_{\theta\theta} = \frac{d\mathbf{z}}{d\theta} \cdot \frac{d\mathbf{z}}{d\theta} = \|\frac{d\mathbf{z}}{d\mathbf{x}} \frac{d\mathbf{x}}{d\theta}\|^2$ where $\frac{d\mathbf{z}}{d\mathbf{x}} = W\mathrm{diag}(\phi'(W[\cos(\theta), \sin(\theta)]))$ and $\frac{d\mathbf{x}}{d\theta} = [-\sin(\theta), \cos(\theta)]$, for a hidden layer activation $\mathbf{z} = \phi(W\mathbf{x})$ and an input $\mathbf{x} = [\cos(\theta), \sin(\theta)]$. Visualising the pullback metric as gridlines on the input manifold by integrating $\sqrt{G_{\theta\theta}}$ over $\theta$ highlights that space was warped (stretched) around the decision boundaries on the circle (Fig. 2c). This means that near the decision boundary, small changes in the input angle lead to large changes in the internal representation — corresponding to changes from one binary output to the other. In comparison, the untrained network did not show this effect (S8.1).

Figure 2: **The pullback metric captures features of task-computation. a.** Network with one hidden layer and tanh nonlinearity trained to map $\mathbf{x} = [\cos(\theta), \sin(\theta)]$ to $\mathbf{y} = \mathbb{1}_{\theta \in [0,\pi)} - \mathbb{1}_{\theta \in (\pi, 2\pi)}$. **b.** Activation of the three hidden units of the network in response to the inputs. **c.** Metric learned by the network — represented as gridlines on the input manifold — illustrating that the manifold has been warped around the class boundary.

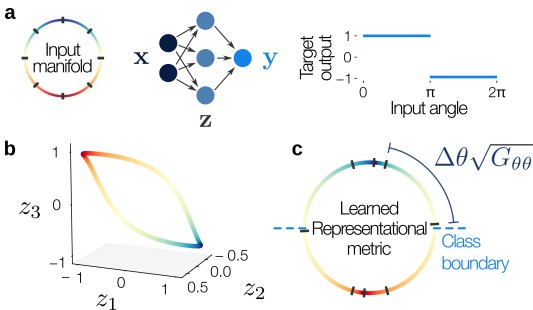

Hence, the computations performed by a deep neural network can be understood via the warping of the internal representations of its input variables. The next section generalises this to dynamical systems, where the inputs, and therefore the representation itself, are time-varying.

# 3 Dynamical systems are constrained by their input topology

**Dynamic neural representations.** In the static network considered above, the activation of the hidden units was naturally constrained to the same manifold as the input itself. Thus, the topology of the input manifold places strong constraints on the hidden-layer topology. We next generalise this relationship to that between the manifold of time-varying inputs to a dynamical system and the manifold to which its state is constrained. Throughout, we consider a dynamical system of the form:

$$\dot{\mathbf{x}}(t) = \mathbf{f}(\mathbf{x}(t), \mathbf{u}(t)), \quad \mathbf{x}(0) = \mathbf{x}_0 \in \mathbb{R}^n, \quad \mathbf{x}(t) \in \mathbb{R}^n, \quad \mathbf{f} : \mathbb{R}^n \times \mathbb{R}^d \to \mathbb{R}^n$$

Our central theorem says that if the inputs $\mathbf{u}$ lie on a low-dimensional manifold then so does $\mathbf{x}(t)$.

**Theorem 3.1.** *Consider a dynamical system as above, and let $\mathbf{u} \in \mathcal{M}$ where $\mathcal{M}$ is an $m$-dimensional manifold. Then $\mathbf{x}(t) \in \mathcal{N}$ where $\mathcal{N} = P(\mathcal{M} \times \mathbb{R})$ and $P$ is a projection map.*

This theorem explicitly characterises the dimensionality of the manifold of states of the system.

**Corollary 3.2.** *If $\mathcal{M}$ is $m$-dimensional then $\mathcal{N}$ is at most $m + 1$-dimensional.*

We stress that the dimensionality $m$ of the input manifold need not match the dimensionality $d$ of the space in which the inputs lie at any particular time point. For example, constant inputs of varying angles might be embedded as $\mathbf{u}(t) = [\cos(\theta), \sin(\theta)]$ and therefore have $d = 2$ and $m = 1$ (i.e. $m < d$). In contrast, the space of all possible Morse codes is infinite-dimensional ($m = \infty$) yet has $d = 1$ (i.e. $m > d$). Theorem 3.1 shows that $m$ rather than $d$ determines the dimensionality of the dynamical system's manifold. In particular, in many — but not all — RNN models of neuroscience tasks $m$ is as low as 1 or 2 (Table 1).

Table 1: Tasks modelled using RNNs tend to have low $m$ while the $d$ can vary based on modelling choices.

| Task: | Working memory[48] | Decision making[39] | Timing[49] | Navigation[50] |
|---|---|---|---|---|
| $m =$ | 2 | 2 | 1 | $\infty$ |
| $d =$ | 3 | 4 | 2 | 2 |

Theorem 3.2 provides a clear result regarding the dimensionality of the manifold of solutions to a dynamical system. This manifold is also a submanifold of the state-space $\mathbb{R}^n$.

**Corollary 3.3.** *Let $\left\{ \frac{\partial}{\partial \mathbf{u}_i} \right\}_{i \in [m]}$ a basis of $T_{\mathbf{u}}\mathcal{M}$. Define $\varphi$ the solution of the dynamical system:*

$$\varphi(\mathbf{u}, t) = \mathbf{x}_0 + \int_0^t \mathbf{f}(\mathbf{x}, \mathbf{u}(\tau)) d\tau, \quad \varphi : \mathcal{M} \times \mathbb{R}_+ \to \mathbb{R}^n$$

*Then if $\operatorname{span} \left\{ \mathrm{d}\varphi \frac{\partial}{\partial t}, \mathrm{d}\varphi \frac{\partial}{\partial \mathbf{u}_i} \right\}_{i \in [m]} \cong \mathbb{R}^{m+1}$ for all $p$ then $\mathcal{N}$ is an immersed submanifold of $\mathbb{R}^n$.*

Theorem 3.2 and corollary 3.3 are illustrated in figure 3a. Often, one will parametrise the input functions with a real-valued variables $\boldsymbol{\kappa} \in \mathbb{R}^m$ (i.e. in local coordinates of $\mathcal{M}$), with the dynamics then being: $\dot{\mathbf{x}} = \mathbf{f}(\mathbf{x}, \mathbf{u}_{\boldsymbol{\kappa}}(t))$. Then, this basis can be more concretely written as $\frac{\partial \mathbf{x}}{\partial \kappa_i} = \frac{\partial \mathbf{x}}{\partial \mathbf{u}} \frac{\partial \mathbf{u}}{\partial \kappa_i} \in \mathbb{R}^n$. Our numerical analyses will depend on this basis and the metric resulting from it. In supplementary materials S7 we extend this result to: i) the case where the dynamical system is defined on another space than the Euclidean space ii) the case where the initial state of the system also varies on a low-dimensional manifold iii) the effect of stochasticity in the dynamics. Similarly to the static case, a natural metric can be defined on this manifold characterised via an adjoint dynamical system:

**Theorem 3.4.** *In local coordinates, the pullback metric on $\mathcal{N}$ is $G_{\mathcal{N}} = UU^T$ with $U = [[\mathbf{f}, 1, \mathbf{0}], [\mathbf{a}, \mathbf{0}, I]]$ where $\mathbf{a}_i = \frac{\partial \mathbf{x}}{\partial \kappa_i} \in \mathbb{R}^n$ follows the dynamics[1]:*

$$\dot{\mathbf{a}}_i = J_{\mathbf{f}} \mathbf{a}_i + \mathbf{h}(\mathbf{x}, t, \kappa)$$

where $J_{\mathbf{f}}$ is the Jacobian of $\mathbf{f}$ and $\mathbf{h}$ a control term. A case that will be of particular interest to the rest of this paper is that of RNNs. In this case the metric is given by the following theorem.

**Corollary 3.5.** *Consider the RNN dynamical system: $\dot{\mathbf{x}} = \mathbf{f}(\mathbf{x}, \mathbf{u}(t)) = W\phi(\mathbf{x}) - \mathbf{x} + B\mathbf{u}(t)$ with $\mathbf{u} \in \mathcal{M}$, then the metric on the RNN manifold in a particular coordinate chart is given by:*

$$G_{RNN} = \begin{bmatrix} ||\mathbf{f}||^2 & \mathbf{f}^T A \\ A^T \mathbf{f} & A^T A \end{bmatrix}, \quad \dot{A} = (W\phi'(\mathbf{x}) - I)A + B\frac{d\mathbf{u}_{\boldsymbol{\kappa}}}{d\boldsymbol{\kappa}}(t), \quad A(0) = 0 \in \mathbb{R}^{n \times m}$$

---

[1]From here on, we ignore time dependencies for notational clarity.

**Low-rank RNNs constrain the embedding dimensionality of the manifold.** Recent work has focused on studying dynamical systems with low-rank weights.[51] In particular, it has been shown that the state of low-rank RNNs is naturally constrained to a low-dimensional space.[52] Indeed, if the weights of an RNN are rank $r$, meaning that $W = \sum_{i=1}^{r} \boldsymbol{\alpha}_i \boldsymbol{\beta}_i^T$, the dynamics of the RNN upon receiving a $d$-dimensional input $\mathbf{u}(t)$ can be written as $\dot{\mathbf{x}} = -\mathbf{x} + \sum_{i=1}^{r} \boldsymbol{\alpha}_i (\phi(\mathbf{x}) \cdot \boldsymbol{\beta}_i) + \sum_{i=1}^{d} \mathbf{b}_i u_i(t)$ which is a linear combination of the $\boldsymbol{\alpha}'s$ and $\mathbf{b}$'s (plus a leak term). This means that the state of the system will be mostly constrained to the $r + d$-dimensional subspace spanned by the $\boldsymbol{\alpha}_i$'s and $\mathbf{b}_i$'s, with rapidly decaying dynamics in all other directions. Thus, the low-rank RNN framework places constraints on the linear embedding dimensionality of the state while our framework places constraints on the intrinsic dimensionality of the nonlinear manifold of states. The two frameworks could naturally be combined in order to design RNNs with specific intrinsic and embedding dimensionalities, by choosing the proper input manifold and rank of the weights.

**Attractors have rank-deficient metrics.** Another common geometric perspective on dynamical systems comes from studying *attractor* manifolds. Here we show how the metric can be used to understand the computational mechanisms behind a classic attractor RNN model.[53] This RNN consists of three units receiving two constant-in-time inputs (Fig. 3b). The two inputs are correlated, such that the set of all possible inputs forms a line manifold. Starting at a fixed initial state the network draws a trajectory which depends on the value taken by the input before settling on a line of fixed points (Fig. 3c). At time $t = 0$, the state of the system is independent of the input and therefore the metric is rank-deficient with a single non-zero entry given by $G_{tt} = \mathbf{f}(\mathbf{x}, \mathbf{u}(0))^T \mathbf{f}(\mathbf{x}, \mathbf{u}(0))$ where $\dot{\mathbf{x}} = \mathbf{f}(\mathbf{x}, \mathbf{u}(t)) = W\phi(\mathbf{x}) - \mathbf{x} + \mathbf{u}(t)$ and zero entries $0 = G_{tu} = G_{ut} = \mathbf{a}(0)^T \cdot \mathbf{f} = \mathbf{0} \cdot \mathbf{f}$ and $0 = G_{uu} = \mathbf{a}(0)^T \cdot \mathbf{a}(0) = \mathbf{0} \cdot \mathbf{0}$. When $t \to \infty$, because of the way the weight matrix $W$ is chosen, the system converges to steady-state meaning that $G_{tt} \to 0$. However, the state to which is converges is dependent on the value taken by the input and therefore $G_{uu} > 0$ (Fig. 3d).

Thus, dynamical systems lie on low-dimensional manifolds whose topology is constrained by their inputs and whose geometry reflects the computations they perform. Over the next sections we show how this provides novel insights into classic RNN models.

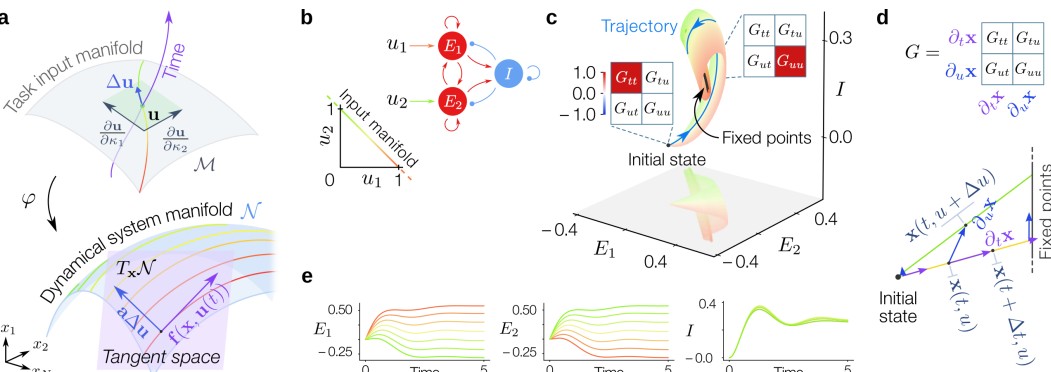

Figure 3: **Dynamical systems' states lie on manifolds whose geometry capture task computations.** **a.** Schematic of how a manifold of time-varying inputs generates a dynamical system manifold. Each point $\mathbf{u}$ on the input manifold $\mathcal{M}$ is a time-varying function. The integral function $\varphi$ maps this input function to the time-varying solution to the dynamical system. The tangent space of the dynamical system manifold is thus spanned by two vectors corresponding to small changes in the input parameters $\boldsymbol{\kappa}$ (given by the adjoint $\mathbf{a}$) and small changes in the time point $t$ (given by $\mathbf{f}$). **b.** E-I network of decision making, whose constant-in-time inputs lie on a line manifold. **c.** The metric captures the transition from the initial state to a line of attractor fixed points. **d.** The metric is the inner product between the tangent vectors. **e.** Single-neuron activity across different trajectories.

## 4 Contextual inputs warp neural manifolds

Biological and artificial agents must flexibly adapt to changes in task context to focus on relevant information while ignoring irrelevant inputs. Contextual evidence integration is a classic neuroscience task which has been modelled extensively using RNNs.[23,54,55] Previous work has analysed this task by studying local linearised dynamics about fixed points.[39] In particular, it has been suggested that

RNNs solve the task via line attractor manifolds. Here, we provide a fully nonlinear analysis of the dynamics and geometry of the computations underlying the solution to this task in RNNs. In particular, we show that the activity lies on a separate three-dimensional manifold for each context and that, over time, the manifold warps to compress the irrelevant information.

**Stochastic differential equations model noisy evidence integration.** We trained an RNN on a classic contextual binary evidence integration task with dynamics:

$$\mathbf{dx} = (W\phi(\mathbf{x}) - \mathbf{x} + \mathbf{b}_1 u_1 + \mathbf{b}_2 u_2 + \mathbf{c}_1 \mathbb{1}_{ctx=1} + \mathbf{c}_2 \mathbb{1}_{ctx=2})dt + B d\mathcal{W}, \quad \mathbf{x}(0) = \mathbf{0}$$

where $\mathbf{x}(t) \in \mathbb{R}^n$ and $B = [\mathbf{b}_1, \mathbf{b}_2]$ such that $d\mathcal{W} \in \mathbb{R}^{2 \times n}$ is a Wiener process increment. In each context ctx $= i$, the network is trained to output the sign of the relevant input $u_i$ at the final time point $t = 10$:

$$l_{ctx}(W, A, D) = ||y(t = 10) - \text{sign}(u_1 \mathbb{1}_{\text{ctx}=1} + u_2 \mathbb{1}_{\text{ctx}=2})||^2, \quad \text{ctx} \in \{1, 2\}, u_i \in [-0.2, 0.2]$$

where $y(t) = D\phi(\mathbf{x}(t))$ and $D \in \mathbb{R}^{1 \times n}$ is a decoder. Thus, the network must learn to discard the irrelevant input in each context. The input manifold is generated by varying the two inputs $(u_1, u_2)$ for a fixed realisation of the noise (Fig. 4**a**; *right*).

**The network solves the task by suppressing output variability in response to irrelevant inputs.** For a given context, the input naturally lies on a two-dimensional manifold defined by the coordinates $u_1$ and $u_2$ (Fig. 4**a** *left*). Thus, the neural activity lies on a three-dimensional manifold generated by the two input dimensions and the time direction. Furthermore, switching between contexts generates two such manifolds (Fig. 4**b**). As in the previous section, the tangent space of this manifold has a natural basis given by $\partial_t \mathbf{x}, \partial_{u_1} \mathbf{x}, \partial_{u_2} \mathbf{x}$. To understand how the network processed relevant and irrelevant information in each context, we asked how the readout of the network $\mathbf{y}$ changed when varying the state of the network along each basis vector. We found that, as time progresses, varying the relevant, but not irrelevant, input leads to a large change in the readout (Fig. 4**c**). We further illustrate this by plotting the output of the network under a fixed realisation of the noise process while varying the irrelevant input (Fig. 4**d**).

**The metric warps the manifold to discard irrelevant inputs.** Thus, the trained network is able to flexibly discard irrelevant information in each context. We next sought to understand the geometry behind this computation. To this end, we visualised the manifold at different time points in a two-dimensional subspace spanned by two neurons. We found that as time progresses, the direction on the manifold corresponding to the irrelevant input was compressed (Fig. 4**e**). To verify this insight at the network level, we computed the metric over the three intrinsic dimensions of the manifold. We found that, initially, the metric's diagonal entries were roughly equal, suggesting an equal sensitivity to changes in the two inputs and the time dimension. In contrast, near the readout time, i) the time component of the metric had converged to near zero, suggesting that the network had reached steady state, and ii) depending on the context, the component of the metric corresponding to the irrelevant input had significantly decreased, suggesting that the irrelevant input no longer influenced the state of the network (Fig. 4**e**). This confirmed that, as time progresses, the neural manifold is compressed along the irrelevant input direction. To visualize this, we computed geodesics under the metric at early and late time points, which confirmed that space had been warped on the manifold, both by compressing the irrelevant direction and by stretching the manifold near the decision boundary along the relevant direction (Fig. 4**f**). Moreover, as time progressed, two of the eigenvalues of the metric — corresponding to the time and irrelevant input component — decayed to zero (Fig. 4g), suggesting that the manifold had become closer to one-dimensional by the decision time.

**The basis of the tangent space provides insight into neural connectivity and responses.** Finally, we sought to relate these geometric results to the connectivity and dynamics of the network. Consistent with recent work suggesting that RNNs trained on low-dimensional tasks have low-rank connectivity,[23,55] we found that the changes in weights between the first and last training iteration $\Delta W = W_K - W_0$ (and the weights themselves, S8.3) were low-rank (Fig. 4**h**). This suggested that the dynamics of the system could be understood in terms of a small number of dynamical modes.

Classic work based on linear(ised) models of the task has argued that context-dependent alignment of dynamical modes, or "selection vectors" (left eigenvectors of the weight matrix), with relevant inputs underlies the solution to the task.[39] We asked whether a geometric analysis of the manifold could reveal a similar mechanism. Indeed, we found that the tangent vectors corresponding to each input were initially both partially aligned to the leading right singular vector of the weight matrix, but over time the alignment of the relevant input's tangent vector increased while the irrelevant input's tangent

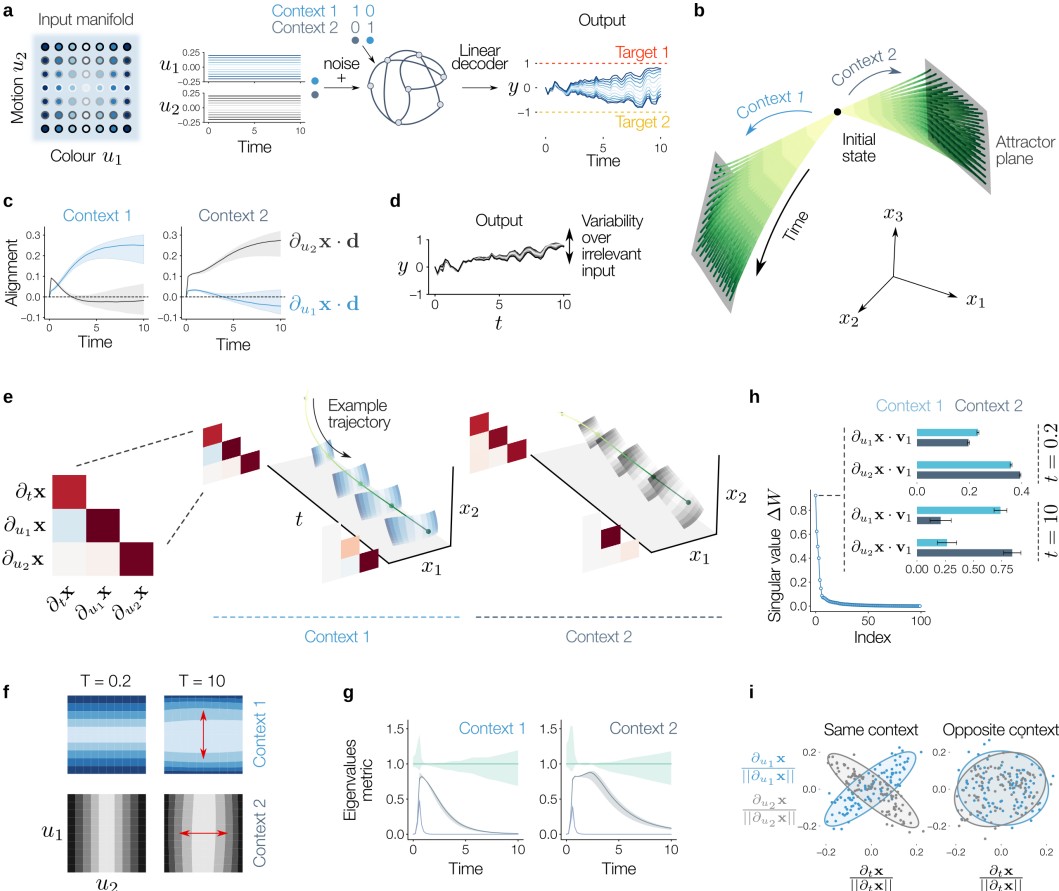

Figure 4: **Neural manifold warping of irrelevant inputs during contextual evidence integration.** **a.** Task schematic: the task consists of two noisy inputs at different expected magnitude. Depending on the contextual cue, the network has to output the sign of the relevant input. Right: example outputs under the same noise realisation across different magnitudes of the relevant input. **b.** Because the input lies on a 2D manifold, neural activity lies on a 3D manifold (here shown in the space of three neurons). Further, there are two manifolds, one for each context. These manifolds are bounded by planes of attractor fixed points (see panel e). **c.** Changes in the relevant (but not irrelevant) input lead to changes in the decoder output at late (but not early) time points. Error bars are across the irrelevant input. **d.** This ensures that the output is not affected by variability in the irrelevant input. **e.** 2D time-slices of the manifold. The metric goes from being everywhere diagonal with equal diagonal entries at early time points to having a near-zero time component (because the network has converged to an attractor state) and a dominant input space component that depends on the context. **f.** Geodesic gridlines under the pull-back of the metric at two different time points highlights that space becomes stretched near the decision boundary along the relevant input **g.** Eigenvalue of the metric (normalised to 1, error bars across the irrelevant input). The manifold becomes pseudo-Riemannian at large times, such that only the largest eigenvalue (corresponding to the relevant input) is large. **h.** Alignment of the largest eigenvector of the weight updates with each basis vector of the tangent space. Over time the largest eigenvector becomes more aligned with the relevant input. **i.** The change in activation of neurons following a change along the each basis vector of the tangent space follow a Gaussian distribution. The covariance of this distribution is correlated/anti-correlated between the time- and relevant-input but not between the time- and irrelevant-input.

vector decreased (Fig. 4**h**). Thus, context-dependent alignment of recurrent dynamical modes can be captured through an analysis of the low-dimensional nonlinear geometry of the system, without linearisation around fixed point manifolds.

Finally, to understand how individual neurons participated in these low-dimensional dynamics, we asked how their responses were distributed with respect to the tangent space of the manifold.

Individual neurons' responses were approximately Gaussian distributed along the tangent vectors, suggesting a highly distributed embedding of the low-dimensional manifold in neural state space (Fig. 4i). Furthermore, neurons participated in the relevant input and time directions in a correlated manner, while participation in the irrelevant input and time directions was uncorrelated, suggesting a highly structured embedding of the relevant input dimension of the neural manifold in state space together with a disordered embedding of the irrelevant input dimension.

Overall, our analysis suggests that dynamical neural representations can flexibly warp based on contextual cues to discard irrelevant information.

## 5 Working memory uses a hyper-torus with dynamic geometry

In the previous section we saw that contextual cues can warp the manifold to discard irrelevant inputs. Here we hypothesize that this dynamic warping mechanism can be used to encode and retrieve a sequence of inputs. Recent studies have characterised the neural geometry of biological[56,57] and artificial[48,58–60] networks performing "working memory" tasks in which information is stored in the activation of a network and retrieved at a later time point. Sequentially presented stimuli have been suggested to be stored in orthogonal subspaces in neural activity.[56] In addition, neural representations of stimuli drawn from a continuous input manifold have been shown to be warped into a set of discrete attractors.[58,61] However, how neural geometries encoding multiple stimuli are dynamically reoriented or warped during the encoding, delay and retrieval of working memories remains unclear. At two extremes, stimuli encoded in orthogonal subspaces may be retrieved sequentially by dynamically realigning each subspace to a fixed decoder, or the geometry of the neural representation may be dynamically warped to increase the projection of each subspace onto the decoder (Fig. 5).

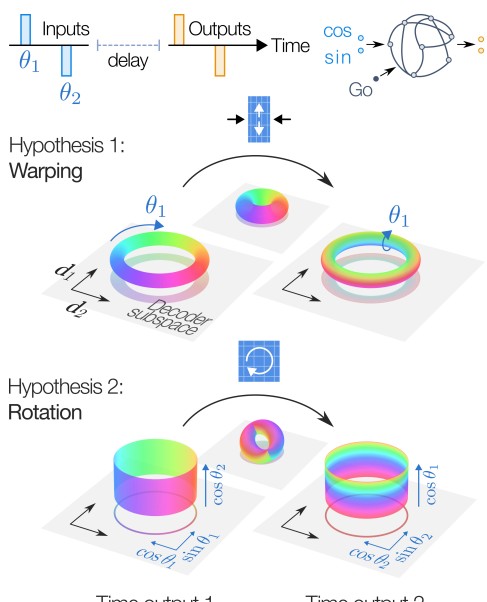

**RNNs solve the task using a curved geometry.** We tested these hypotheses in RNNs trained to perform a sequential working memory task. The RNN sequentially received two inputs drawn from a circular manifold, and was required to output the same sequence after a variable delay (Fig. 5 *top*):

Figure 5: Task where a network has to remember two sequential inputs and then output them after a variable delay. Possible task solutions: **i) dynamically warping the torus** to compress the irrelevant input's representation or **ii) realigning the torus'** relevant input encoding direction to the decoder subspace while preserving a fixed intrinsic geometry.

$$\dot{\mathbf{x}} = W\phi(\mathbf{x}) - \mathbf{x} + B\mathbf{u}(t) + \mathrm{go}_h(t)\mathbf{c}, \quad L(W, A, D) = \int_0^{T-h} ||D\phi(\mathbf{x}(t+h)) - \mathbf{u}(t)||^2 dt$$

with $\mathbf{u}(t) = [\cos\theta(t), \sin\theta(t)]$ where $\theta(t)$ was piece-wise constant, and $h \in [0.5, 3.5]$ was the delay period, ended by a pulse $\mathrm{go}_h(t)$. Applying PCA to the activity at different points revealed a toroidal manifold with dynamic geometry (Fig. 6**a-b**). Yet this low-dimensional visualisation is insufficient to fully characterise the RNN's representational geometry, which requires analysis of the metric.

**The metric dynamically warps the torus to represent time-specific relevant information.** To test for warping of the neural manifold we computed the Gaussian curvature of the torus at different time points (Fig. 6**c**; S8.4). We found that the torus had a non-flat geometry, including both positive and negative curvature which was highly non-uniform over the torus (Fig. 6**c-d**). Furthermore, the metric revealed that: i) during the input period the torus' shape was formed ii) over the delay it remained stable and iii) during the retrieval phase the RNN selectively compressed the stimuli not immediately retrieved (Fig. 6**e**). Finally, we looked at how these changes in intrinsic geometry related to the extrinsic embedding of the torus in the state space. First, the decoder was selectively aligned to the basis vector encoding the relevant stimulus at each time point during retrieval (Fig. 6**f**) and the subspace spanned by the top two principal components was stable across the delay (Fig. 6**g**).

**Storing more memories requires higher-dimensional manifolds.** Finally, we ask how these results generalise to storing more memories. First, we proved (S8.4) that the activity of the RNN during the delay period lies on a hyper-torus whose dimension equals the number of items stored, and that, furthermore, to optimally encode the stimuli this torus must be embedded in the state space (as opposed to immersed). Computing the geodesic distance on the hyper-torus w.r.t. a reference combination of angles revealed how the hyper-torus was compressed along irrelevant directions at different time points to retrieve different memories (Fig. 6h-i).

In sum, we mathematically show that sequential working memory relies on higher-dimensional representations. In RNNs, a combination of dynamical changes in the intrinsic and embedding geometry of the manifold occurs during retrieval of these memories.

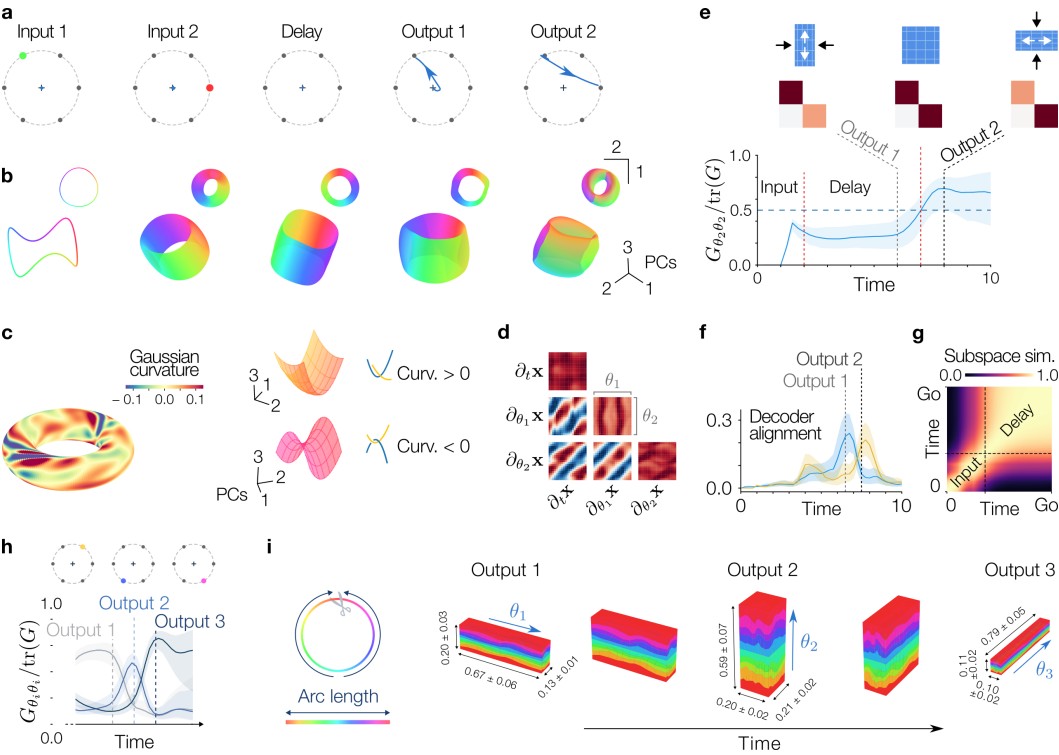

Figure 6: **Recalling memories requires dynamically warping their neural representations.**
**a.** Example of network input and output. **b.** Different time slices of the RNN manifold, shown projected on the first three principal components (PCs) (inset two PCs). After the first input comes on the state at any particular time point is lies on a 1D manifold encoding the input; after the second input comes on the state is constrained to a torus encoding both inputs. **c.** Gaussian curvature of the torus at output onset ($t = 6$). The torus has positive and negative curvatures at different points. *Right:* Two different pieces of the torus projected in the PC space displaying positive and negative curvatures at the start of the delay period ($t = 2$). **d.** Metric of the RNN manifold at the start of the delay period ($t = 2$). **e.** Average stretching of the torus, quantified by the ratio of one of the diagonal entries of the metric to its trace. *Top:* Metric averaged over the torus. The torus goes from being warped along the encoding of the first then second angle during retrieval. **f.** Alignment of the basis of the tangent space to the decoder $||D\text{diag}(\phi'(\mathbf{x}))\partial_{\theta_i}\mathbf{x}||$. Blue is $\theta_1$ and yellow is $\theta_2$. **g.** Subspace similarity between the spaces spanned by the first two PCs as computed at any pair of time points. **h.** Ratio of individual entries of the diagonal of the metric to its trace. **i.** Arc length from a reference angle. The hyper-torus is stretched dynamically through the recall period to compress the two irrelevant angles.

# 6 Discussion

**Conclusion.** In this work, we developed a mathematical framework extending previous applications of Riemannian geometry in machine learning to dynamical systems receiving time-varying inputs on low-dimensional manifolds. By deriving the pullback Riemannian metric on the manifold of system states, we showed that the geometry of a dynamical system reflects its computations, linking representational geometry to computation-through-dynamics. This framework thus provides a mathematical description of how dynamical systems transform input manifolds into neural activity manifolds whose topology and geometry are constrained by the structure of their inputs.

In Section 3, we used this framework to show that contextual inputs warp the geometry of neural manifolds in RNNs trained on decision-making tasks. Specifically, we found that the metric compresses directions corresponding to irrelevant inputs while preserving those corresponding to relevant task variables, enabling flexible context-dependent computations. In Section 4, we extended this analysis to sequential working-memory tasks and showed that RNNs encode and retrieve multiple memories through dynamic warping of a higher-dimensional manifold, specifically a hyper-torus whose intrinsic and embedding geometry evolve over time. Together, these results demonstrate that warping is a ubiquitous feature of the computations performed by RNNs across distinct task domains.

**Limitations.** The relationship between geometry, dynamics and connectivity is degenerate: the same geometry (i.e. the same metric) can arise through different network architectures. Future work could investigate whether networks with identical geometries but distinct connectivity and dynamics can implement distinct classes of computations, or instead constitute a degenerate class of solutions to the same problem. A second limitation is our analysis of the influence of noise on geometry. In particular, we treated noise in a *pathwise* manner; we studied the manifold generated under a fixed realisation of the noise process. The distribution of manifolds generated by random noise realisations remains to be characterised, and could provide insight into the geometry of computations on stochastic data.

**Broader impacts.** Our framework is applicable to a wide range of dynamical systems. In particular, methods to infer dynamical systems from biological or physical data using machine learning tools have been widely adopted, and the Riemannian geometry of these systems could analysed to interpret the behaviour of these data-driven dynamical systems. In particular, the computational mechanisms identified here in task-trained RNNs could be investigated in RNNs fit directly to neural data in order to test hypotheses regarding the geometry of computation in large-scale neural recordings.

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

# Supplementary Materials

## S7 The representational metric of dynamical systems

In this section we introduce the mathematical framework used throughout the main manuscript. In order for this work to be self-contained, we first briefly review the relevant concepts of differential and Riemannian geometry. To provide intuition in a mathematically simplified setting, we study deep neural networks as a series of geometric transformations, showing that this view can provide insights into the computations performed by the network. We then provide our main contribution: a full derivation of the geometry of dynamical systems' manifolds. Finally, we show a concrete application of these mathematical results for continuous-time RNNs.

### S7.1 Differential geometry of deep neural networks

The results of this subsection mostly follow from elementary differential geometry (e.g. we refer the reader to[1]), and we therefore state them without derivation.

**The topological and differentiable structure of deep neural networks.** In this section we consider a generic depth-$l$ neural network of the form:

$$\varphi_l = f_l \circ f_{l-1} \circ ... \circ f_1 : \mathbb{R}^{n_0} \to \mathbb{R}^{n_l}$$

where $f_k$ is the map from hidden layer $k-1$ to hidden layer $k$. For example, $f_k(\mathbf{z}_{k-1}) = \tanh(W_k\mathbf{z}_{k-1} + \mathbf{b}) = \mathbf{z}_k$ where $\mathbf{z}_k \in \mathbb{R}^{n_k}$ is the vector of activations of the neurons in the $k$th hidden layer. Following previous work,[2] we consider the input to this network as living on a low-dimensional manifold:

$$\varphi_k(\psi(p)) = \varphi_k(\mathbf{x}) = \mathbf{z}_k, \quad p \in \mathcal{M}$$

where $\psi$ is an embedding of $\mathcal{M}$ in $\mathbb{R}^{n_0}$. The following proposition gives a clear characterisation of the topology of the manifold over which $\mathbf{z}$ lies. In particular, if $\varphi_k : \mathbb{R}^{n_0} \to \mathbb{R}^{n_k}$ is i) continuous ii) invertible and iii) its inverse is continuous then it is a *homeomorphism*.

**Proposition S7.1.** *Let $\mathbf{x} \in \mathbb{R}^{n_0}$ be constrained to $\psi(\mathcal{M})$, an embedding of $\mathcal{M}$ in $\mathbb{R}^{n_0}$, and let $\varphi_k$ be as defined above. If $\varphi_k$ is a homeomorphism then $\mathbf{z}_k = \varphi_k(\mathbf{x})$ is also constrained to $\mathcal{M}$.*

Importantly, $\varphi_k$ is a simple real function, and its invertibility and continuity can be verified via basic calculus. Thus, $\mathbf{x}$ and $\mathbf{z}$ are on the same manifold, simply represented differently in $\mathbb{R}^{n_0}$ and $\mathbb{R}^{n_k}$.

In general, the invertibility and continuity of the network transformation are mostly independent of the parameters of the network. For example, if the network is of the form $f_k = \phi(W_k\mathbf{x} + \mathbf{b})$ where $\phi$ is an element-wise nonlinearity, then if $\phi$ is (element-wise) homeomorphic (e.g. tanh, softplus, leaky-ReLU, but not ReLU), and $W_k$ is full rank, and the network doesn't have a bottleneck layer, then these conditions are satisfied. In particular, $W_k$ is full rank with probability 1 if it is initialised with entry-wise i.i.d. Gaussian or uniform weights. Thus, provided the rank of $W_k$ does not change over training, the topology of a trained network will match that at initialisation, and hence does not reflect the computations learned by the network. However, in some instances the topology may indeed change: recent work has highlighted the ubiquity of low-rank networks,[3] and depending on the objective, the learned weights may be of a rank lower than the minimum embedding dimensionality of the manifold — trivially one could think of a network trained to map all its inputs to 0. Nevertheless, in general it is the geometry of the neural representation that provides most insight into the computations learned by the network, as this geometry depends continuously on the parameters of the network whereas the topology is invariant to large classes of parameter changes.

There are multiple routes to characterising the geometry of a manifold. Here we will specifically consider *smooth manifolds*. In particular, such smooth manifolds have a tangent space $T_p\mathcal{M}$ at any point $p \in \mathcal{M}$.

**Proposition S7.2.** *Let $\varphi_k : \mathbb{R}^{n_0} \to \mathbb{R}^{n_k}$ differentiable with Jacobian $J_k \in \mathbb{R}^{n_k \times n_0}$. If $J_k$ is full matrix rank and $\psi$ is an embedding, then $d(\varphi_k \circ \psi) = J_k d\psi(\mathbf{v}) : T_p\mathcal{M} \to T_{\varphi_k(\psi(p))}$ is surjective and $\mathcal{M}$ is an immersed submanifold of $\mathbb{R}^{n_k}$*

We note here that the differentiability and homeomorphicity conditions are overlapping but not restrictions of one another. In particular, even when $\varphi_k$ is not bijective $d\varphi_k$ can still be surjective

and therefore its Jacobian full rank. Similarly, $\varphi_k$ bijective and differentiable does not imply $\mathrm{d}\varphi_k$ surjective.

**Riemannian geometry provides insights into network computations.** So far the structure we've explored had no geometry, in that we did not define a notion of distance between points on the manifold. For this, we must add a *metric* to the manifold.

**Definition S7.1.** A *metric* $g$ is a function defining the inner product between vectors of the tangent space of a manifold. The metric is Riemannian if it satisfies: $g : T_p\mathcal{M} \times T_p\mathcal{M} \to \mathbb{R}$ such that $g(\mathbf{v}, \mathbf{v}) > 0$ if $\mathbf{v} \neq \mathbf{0} \in T_p\mathcal{M}$ and $g(\mathbf{v}_1, \mathbf{v}_2) = g(\mathbf{v}_2, \mathbf{v}_1)$

Importantly such a metric allows defining the geodesic distance between points:

**Definition S7.2.** Let $\gamma : [0, T] \to \mathcal{M}$ a smooth curve on $\mathcal{M}$ where $T \in \mathbb{R}$. In particular, this implies that $\dot{\gamma}(t) \in T_{\gamma(t)}\mathcal{M}$, and we can define the *length* of $\gamma$ as:

$$L(\gamma) = \int_0^T g(\dot{\gamma}(t), \dot{\gamma}(t))dt.$$

Furthermore, the geodesic distance between $p_1 \in \mathcal{M}$ and $p_2 \in \mathcal{M}$ is defined as:

$$d(p_1, p_2) = \inf_\gamma L(\gamma), \quad \text{constraining } \gamma(0) = p_1, \gamma(T) = p_2$$

This definition was central to our study of dynamical systems in the main text.

In figure 2 the manifold considered was the circle $\mathcal{M} = S^1$ and representing different points on the circle via their angle ($p = \theta$), the embedding of the circle in the 2-d input space was $\mathbf{x} = \psi(\theta) = [\cos(\theta), \sin(\theta)]$. However, we could have chosen a different embedding, say $\mathbf{x} = [(1 + 0.1\sin(\theta))\cos(\theta), \sin(\theta)]$, which in general will affect the geometry of the hidden representation $\mathbf{z}_k$. Indeed, suppose the target output is independent of the input geometry — as is the case in figure 2 where it is only the point on the circle that determines the output — then the network must perform different computations on these different input representations to perform the same output.

To characterise these transformations, we can *pull back* the usual metric defined on $\mathbb{R}^{n_k}$, which is the standard dot product, to the manifold $\mathcal{M}$.

**Definition S7.3.** The *pullback metric* is defined as:

$$g(\mathbf{v}_1, \mathbf{v}_2) = (\mathrm{d}(\varphi_k \circ \psi)(\mathbf{v}_1)) \cdot (\mathrm{d}(\varphi_k \circ \psi)(\mathbf{v}_2)), \quad \mathbf{v}_1, \mathbf{v}_2 \in T_p\mathcal{M}$$

in particular $\mathrm{d}(\varphi_k \circ \psi)(\mathbf{v}_i) \in \mathbb{R}^{n_k}$ and $\cdot$ is the standard Euclidean dot product.

Thus, we can define a metric on $\mathcal{M}$ by taking tangent vectors in $T_p\mathcal{M}$, pushing them forward through the network, where at the $k$th layer they become vectors of $T_p\mathbb{R}^{n_k} = \mathbb{R}^{n_k}$, such that we can then simply take their dot product. The value of this dot product is our definition of the inner product between the original tangent vectors in $\mathcal{M}$. The pullback metric represents how the tangent spaces of the manifold are transformed through the layers of the network and therefore the computations of the network. In particular we can more explicitly write the pullback metric as:

$$g(\mathbf{v}_1, \mathbf{v}_2) = (J_k \mathrm{d}\psi(\mathbf{v}_1)) \cdot (J_k \mathrm{d}\psi(\mathbf{v}_2))$$

with

$$J_k = W_k \mathrm{diag}(\phi'(\mathbf{z}_{k-1}))...W_1 \mathrm{diag}(\phi'(\mathbf{x}))\mathrm{d}\psi(\mathbf{v}_1)$$

where $J_k \in \mathbb{R}^{n_k \times n_0}$ is the Jacobian of $\varphi_k$. This highlights the earlier claim that the geometry, as defined via the pullback metric, is directly dependent on the parameters $W_1, ..., W_l$ of the network.

**Geodesic gridlines and computations.** In our networks the input metric was uniform, but in general data of the world maybe be unevenly distributed on the manifold, which will affect the learned parameters, and therefore the learned representation. In the case of our simple network the loss function was more formally defined as:

$$l = \int_{\mathcal{M}} ||\varphi_l(\psi(\theta)) - \mathbf{y}_{target}||^2 \sqrt{\det(h)}\mathrm{d}\theta$$

where $h : T_p\mathcal{M} \times T_p\mathcal{M}$ is the metric on the input manifold. That is, some inputs are more heavily weighted in the loss if space is more spread around them. For example, in a probabilistic setting where

instead of integrating over all possible inputs only a finite sample of them is drawn at each iterations, $\sqrt{\det(h)}$ can be seen as a probability distribution (up to normalisation) on the input manifold — so that some input-output pairs are drawn more frequently than others. In our case all angles were equal, so the input metric was uniform:

$$l = c \int_{\mathcal{M}} ||\varphi_l(\psi(\theta)) - \mathbf{y}_{target}||^2 \mathrm{d}\theta$$

for some $c \in \mathbb{R}_+$. But more generally non-uniform metrics on the manifold of input will lead to different learned representations since the loss is different. For example, a class being more frequent in a binary classification task may change the representation: if in figure 2 we had sampled more frequently angles $[0, \pi)$ than $(\pi, 2\pi)$, it would have led to an asymmetric stretching of space on each side of the class boundaries.

Thus, the pullback metric $g$ gives insights into the learned representation, but the learned representation is dependent on the metric of the input manifold $h$. Using the definition of geodesic distance, both $g$ and $h$ can be used to measure the distance between points on the manifold. But what points to choose to measure the distance between? In 2, 4 and 6 we showed that an insightful reflection of the computations done by the network was provided by the computing the geodesic distance using $g$ between points that were evenly apart under the $h$ metric. That is define points $p_1, p_2$ such that $d^h(p_1, p_2) = 1$ and study $d^g(p_1, p_2)$ where $d^h$ and $d^g$ are the geodesic distance functions under the $h$ and $g$ metric respectively.

**Coordinates on the input manifold provide a natural input metric.** So far our treatment of the topology, smooth structure and geometry of deep neural networks has been relatively free of a choice of a coordinate system for $\mathcal{M}$. An $m$-dimensional manifold is defined as a collection of pieces of $\mathbb{R}^m$ glued together via the so-called transition functions. However, different choices of pieces of $\mathbb{R}^m$ can give rise to the same manifold. A trivial example is that of a line manifold: the open interval $(0, 3)$ and $(10, 16)$ are the same manifold, but endowed with different coordinates; there is an explicit diffeomorphism between them given by $F(p) = 10 + 2p$.

In many cases, there is a meaningful coordinate system — at least around a point — on the input manifold. A manifold of images may for example be characterised by the position of an object in the image or its orientation.[4] It is insightful to understand how a neural network represents these different features. When a specific coordinate system is given, a natural matrix representation of the metric can be defined.

**Definition S7.4.** A given coordinate system[2] $\mathbf{x}(p) = (x_1, x_2, ..., x_m)$ where $\mathbf{x} : \mathcal{M} \to \mathbb{R}^m$ induces a natural basis of the tangent space such that $\mathbf{v} \in T_p\mathcal{M}$ can be written as a linear combination of them:[3]

$$\mathbf{v} = \sum_{i=1}^{m} v^i \frac{\partial}{\partial x_1}$$

In this coordinate system inner products are given by:

$$g(\mathbf{v}_1, \mathbf{v}_2) = \sum_{i,j=1}^{m} v_1^i G_{i,j} v_2^i$$

This higlights that the metric in a particular coordinate system can also be written as a matrix $G \in \mathbb{R}^{m \times m}$ and $G_{i,j} = g(\frac{\partial}{\partial x_i}, \frac{\partial}{\partial x_j})$ are its entries. Furthermore, if the manifold is embedded in the Euclidean space:

$$G = J^T J \in \mathbb{R}^{m \times m}, \quad J = \left[ \frac{\partial p}{\partial p_1}, ..., \frac{\partial p}{\partial p_m} \right] \in \mathbb{R}^{m \times m}$$

Back to our deep neural network, the pullback metric changes from layer to layer as:

$$G_k = \mathrm{d}\psi(\mathbf{v})^T J_k^T J_k \mathrm{d}\psi(\mathbf{v}) = \mathrm{d}\psi(\mathbf{v})^T J_{f_1}^T ... J_{f_k}^T J_{f_k} ... J_{f_1} \in \mathbb{R}^{m \times m}$$

---

[2]The reader may be more familiar with the notation $(U, \varphi)$ for a chart, here we've simply written $\mathbf{x} = \varphi$.

[3]This notation defines $\frac{\partial f}{\partial x_1} = (f \circ \gamma)|_{t=1}$ for the smooth curve $\gamma(t)$ defined such that $x_i(\gamma) = (x_1(p), ..., x_i(p) + t, ..., x_m(p))$ and a smooth function $f : \mathcal{M} \to \mathbb{R}$.

where $J_{f_i} \in \mathbb{R}^{n_i \times n_{i-1}}$ is the Jacobian of $f_i$. This means that:

$$G_k = \mathrm{d}(\varphi \circ \psi)(\mathbf{v})^T \mathrm{d}(\varphi \circ \psi)(\mathbf{v}) \in \mathbb{R}^{m \times m}$$

Integrating along a curve on the manifold under the pullback metric in coordinates thus gives:

$$L(\gamma) = \int_0^T g(\dot{\gamma}(t), \dot{\gamma}(t)) dt = \int_0^T \mathrm{d}(\varphi \circ \psi)(\dot{\gamma}(t))^T G_k \mathrm{d}(\varphi \circ \psi)(\dot{\gamma}(t))^T dt$$

Where now $\mathrm{d}(\varphi \circ \psi)(\dot{\gamma}(t))$ is a usual derivative of a real function, and $G_k$ a real matrix, which can both be obtained using standard calculus tools (e.g. auto-differentiation).

## S7.2  Dynamical systems' topology

We now generalise the theory of the previous subsection to dynamical systems, where the inputs are time-varying functions. As we shall see, a manifold of functions naturally gives rise to a manifold of functions that are solutions to the dynamical system. Much of the work will lie in viewing this manifold of solutions as a submanifold of the state space.

**The topology of the input manifold constrains the topology of the solution to a dynamical system.** Here we consider a more general setting than in the main manuscript where the dynamical system is defined on a manifold. That is:

$$\dot{\mathbf{x}}(t) = \mathbf{f}(\mathbf{x}(t), \mathbf{u}(t)), \quad \mathbf{x}(t) \in \mathcal{R}^n, \mathbf{u}(t) \in \mathcal{U}^d$$

Furthermore, consider a manifold $\mathcal{M}$ of input functions. That is $\mathbf{u} \in \mathcal{M}$ and $\mathbf{u}(t) \in \mathcal{U}$ where $\mathcal{U}$ is the ambient space in which the input is embedded at any particular time point. To define such a *manifold of functions* (or more precisely of curves) we rely on the following construction:

**Definition S7.5.** A *manifold of real curves* is a manifold $\mathcal{M}$ such that $(\mathbf{u}, t) \in \mathcal{M} \times \mathbb{R}$ for $\mathbf{u}(t) \in \mathcal{U}$, and $\mathbf{u} \in \mathcal{M}$ is $\mathbf{u} = \pi(\mathbf{u}, t)$ for $\pi$ the projection map.

On a particular chart, the function can be parametrised:

$$\mathbf{u}_{\boldsymbol{\kappa}}(t), \kappa \in \mathbb{R}^m$$

For example, in section 4 of the main manuscript we've considered $\mathbf{u}_{\boldsymbol{\kappa}}(t) = [\kappa_1, \kappa_2, \mathbb{1}_{\mathrm{ctx}=1}, \mathbb{1}_{\mathrm{ctx}=2}]$ where $\kappa_1, \kappa_2$ were the levels motion and colour evidence. This input, fixing one of the two discrete contexts, indeed lies on a 2-dimensional manifold with planar topology as $\boldsymbol{\kappa} \in \mathbb{R}^2$. We formulate a stronger version of the theorem of the main text:

**Theorem S7.3.** *Consider the dynamical system:*

$$\dot{\mathbf{x}}(t) = \mathbf{f}(\mathbf{x}(t), \mathbf{u}(t)), \quad \mathbf{x}(t) \in \mathcal{R}^n, \mathbf{u}(t) \in \mathcal{U}^d, \mathbf{u} \in \mathcal{M}^m$$

*where $\mathbf{f}$ is at least once differentiable in its arguments, then $\mathbf{x} \in \mathcal{N}^{m+1}$ (the curve) and $\mathcal{N}^{m+1}$ has the topology of $\mathcal{M}^m \times \mathbb{R}$.*

where the superscripts indicate the dimensionality of the manifold. Theorem 3.2 of the main text is therefore just a special case of this theorem as the dimensionality of the dynamical system is directly defined by the manifold topology. To prove this theorem we need the following lemma:

**Lemma S7.4.** [1] *Let $\dot{\mathbf{y}} = \mathbf{g}(\mathbf{y})$, $\mathbf{y}(0) = \mathbf{y}_0$ with $\varphi(t, \mathbf{y}_0) = \mathbf{y}_0 + \int_0^t \mathbf{g}(\mathbf{y}(\tau)) d\tau$ the solution of the dynamical system at time $t$ starting at $\mathbf{y}_0$, then if $\mathbf{g}$ is smooth, $\varphi$ is a diffeomorphism.*

We can now prove the main theorem:

*Proof of theorem S7.3.* We prove this in a particular coordinate chart of $\mathcal{M}$ and $\mathcal{R}$. Then, the dynamical system can be reformulated as (by abuse of notation):

$$\dot{\mathbf{x}}(t) = \mathbf{f}(\mathbf{x}(t), \mathbf{u}_{\boldsymbol{\kappa}}(t)), \quad \mathbf{x}(t) \in \mathbb{R}^n, \boldsymbol{\kappa} \in \mathbb{R}^m$$

We can augment this dynamical system by introducing the variable $\mathbf{y} \in \mathbb{R}^n \times \mathbb{R}^m \times \mathbb{R}$ defined as:

$$\varphi(t, \boldsymbol{\kappa}) = \mathbf{y}(t) = [\mathbf{x}(t), t, \boldsymbol{\kappa}]$$

then:

$$\dot{\mathbf{y}} = [\dot{\mathbf{x}}, 1, \mathbf{0}] = [\mathbf{f}(\mathbf{y}_{1:n}, \mathbf{u}_{\mathbf{y}_{n+2:n+m+1}}(y_{n+1})), 1, \mathbf{0}], \quad \mathbf{y}(0) = [\mathbf{x}_0, 0, \boldsymbol{\kappa}]$$

which is an autonomous dynamical system. By the theorem on solution of ODE, if the r.h.s. is at least once differentiable in its argument (which it is since $\mathbf{f}$ is), there exists a unique solution to this dynamical system. Furthermore, by lemma S7.4, $\varphi$ is a diffeomorphism and since $\boldsymbol{\kappa}$ is on an $m$-dimensional space and $t$ in a 1-dimensional, $\mathbf{y}(t)$ is in an $m+1$-dimensional space. More generally $\mathbf{y}$ is an $m+1$-dimensional space of curves of topology $\mathbb{R}^m \times \mathbb{R}$. Finally, $\mathbf{x}$ and $\mathbf{y}$ are simply related by a projection $\mathbf{x}(t) = P(\mathbf{y}(t)) = \mathbf{y}_{1:n}$.

For the sake of completeness we also show that this holds not only over a single chart but over the whole manifold. Consider a smooth transition function $F_{i,j} : \mathbb{R}^m \to \mathbb{R}^m$ from coordinates of chart $i$ to coordinates of chart $j$ of $\mathcal{M}$, and $\varphi^i(t, \boldsymbol{\kappa}^i)$ the solution to the dynamical system on chart $i$. Then $\varphi^j(t, \boldsymbol{\kappa}^j) = \varphi^j(t, J_{F_{i,j}} \boldsymbol{\kappa}^i)$ where $J_{F_{i,j}}$ is the Jacobian of $F_{i,j}$. Since $\varphi$ is a composition of diffeomorphisms it is a diffeomorphism. $\square$

Different conditions on the projection $P$ will give different properties to $\mathbf{x}$, for example if $P$ is homeomorphic then its preserves the topology of the manifold, if $\mathrm{d}P$ is surjective then $\mathbf{x}$ is on $\mathcal{M} \times \mathbb{R}$ an immersed submanifold of $\mathcal{R}^n$.

We stop to address two natural questions regarding edge-cases of this result: 1) *Asymptotic Regimes.* Our result assumes that $t \in \mathbb{R}$ (transient dynamics) rather than the asymptotic limit ($t \to \infty$). While the input-driven manifold usually matches the input dimensionality during transients, the geometry of the asymptotic invariant set can be fundamentally different. For instance, the mapping from inputs to steady-states may not be a diffeomorphism due to bifurcations, and in chaotic regimes, the system may settle onto strange attractors with fractal dimensionality.[5] 2) *Chaos.* A chaotic system affects the geometry of the manifold, not its topology. To see this, consider pulse inputs of varying magnitude $\mathbf{u}(t) = \boldsymbol{\kappa}\delta_{t_0}(t)$. At time $t = t_0 + \Delta t$, the state lies on a 1-dimensional manifold parametrised by $\boldsymbol{\kappa}$, after what it evolves autonomously. If the system has a positive Lyapunov exponent, small perturbations in $\boldsymbol{\kappa}$ result in exponentially diverging trajectories $\mathbf{x}(t)$. Consequently, the tangent vector $\partial_\kappa \mathbf{x}$ will grow rapidly, and the associated metric tensor entries $G_{\kappa,t}$ and $G_{\kappa,\kappa}$ will diverge. However, despite this warping, the topological result holds.

So far — and in the main text — we have mainly considered fixed initial conditions of the dynamical system but the previous proof naturally gives the following corollary.

**Corollary S7.5.** *If $\mathbf{x}_0 \in \mathcal{Q}^q$ and $\mathbf{u} \in \mathcal{M}^m$ then $\mathbf{y}(t) \in \mathcal{N}^{m+q+1} = \mathcal{M}^m \times \mathbb{R} \times \mathcal{Q}^q$ with $\mathbf{x}(t) = P(\mathbf{y}(t))$ so that varying the initial state only adds extra dimensionalities.*

*Proof.* The proof follows mutatis mutandis from that of theorem S7.3 additionally noticing that if $\mathbf{x}_0 \in \mathcal{Q}^q$ then $\mathbf{y}_0 \in \mathcal{Q}^q \times \{0\} \times \mathbb{R}^m$. $\square$

The effect of noise can also be studied from theorem S7.3. In general, noise processes driving differential equations are sampled from an infinite-dimensional manifold of functions (e.g. $L^2$). Thus, the manifold of states taken by the system under *all* realisations of the noise may in general be the entirety of the state space. However, the following corollary states that fixing the noise realisation generates a manifold:

**Corollary S7.6.** *Consider the stochastic differential equation $\mathbf{dx} = \mathbf{f}(\mathbf{x}, \mathbf{u}(t))dt + \mathbf{g}(\mathbf{x})d\mathcal{W}$, $\mathbf{x}_0 = \mathbf{x}(0)$ where $\mathbf{g}(\mathbf{x}) \in \mathbb{R}^{n \times q}$ is at least once differentiable in $\mathbf{x}$ and $d\mathcal{W} \in \mathbb{R}^q$ are the increments of a Wiener process. Then if $\mathbf{u} \in \mathcal{M}$ then $\mathbf{x}(t) = P(\mathbf{y}(t))$ where $P$ is a projection matrix and $\mathbf{y}(t) \in \mathcal{M} \times \mathbb{R}_+$.*

*Proof.* The proof follows a similar construction as the deterministic case. As in S7.3 let $\mathbf{y}(t) = [\mathbf{x}(t), t, \boldsymbol{\kappa}]$, now $\mathbf{dy} = \bar{\mathbf{f}}(\mathbf{x}, \mathbf{u}_\kappa(t))dt + \bar{\mathbf{g}}(\mathbf{x}, \mathbf{u}_\kappa(t))d\mathcal{W}$ where $\bar{\mathbf{f}} = [\mathbf{f}, 1, \mathbf{0}]$, $\bar{\mathbf{g}} = [\mathbf{g}^T, 0]^T$. Thus $\mathbf{y}$ follows an SDE driven by the same noise realisation. Then, using the smoothness of the solution of an SDE with respect to its initial state,[6] the rest of the proof follow the same steps as that of S7.3. $\square$

This shows that under a fixed noise realisation the state of the system lies on a manifold. In general, for different noise realisations $P$ may yield a submanifold an immersion or an embedding of $\mathcal{M} \times \mathbb{R}_+$. Future work could study the distribution of such objects under the noise, for example characterising the measure of the solutions for which it is a proper embedding or an immersion.

## S7.3  Geometry as input-manifold transformations

Having characterised the geometry and differentiable structure of the manifold of dynamical systems, we next turn to their Riemannian geometry. The main insight will be that the metric on the manifold can be defined in terms of the solution to an adjoint dynamical system. This will provide both theoretical insights and a practical way to obtain the matrix representation of a metric.

*Proof of theorem 3.4.* The Jacobian of $\mathbf{y}$ is given by (ignoring time dependencies):

$$J_{\mathbf{y}} = \begin{bmatrix} \frac{\partial \mathbf{x}}{\partial t} & \frac{\partial \mathbf{x}}{\partial \boldsymbol{\kappa}} \\ \frac{\partial t}{\partial t} & \frac{\partial t}{\partial \boldsymbol{\kappa}} \\ \frac{\partial \boldsymbol{\kappa}}{\partial t} & \frac{\partial \boldsymbol{\kappa}}{\partial \boldsymbol{\kappa}} \end{bmatrix} = \begin{bmatrix} \mathbf{f} & \frac{\partial \mathbf{x}}{\partial \boldsymbol{\kappa}} \\ 1 & \mathbf{0} \\ \mathbf{0} & I \end{bmatrix}$$

where the submatrices are concatenated along their appropriate dimensions. The term $\mathbf{f}$ is simply the r.h.s. of the dynamical system, so it remains to compute the term in blue. Each column of this term is of the form $\frac{\partial \mathbf{x}}{\partial \kappa_i}$, which describes how the state of the system at a particular time point changes for an infinitesimal change in one of the input variables on a particular chart. These are given by:

$$\frac{\partial \mathbf{x}}{\partial \kappa_i} = \frac{\partial}{\partial \kappa_i} \left( \mathbf{x}_0 + \int_0^t \mathbf{f}(\mathbf{x}(t), \tau, \mathbf{u}_{\boldsymbol{\kappa}}(\tau)) d\tau \right)$$
$$= \int_0^t \frac{\partial}{\partial \kappa_i} \mathbf{f}(\mathbf{x}(t), \tau, \mathbf{u}_{\boldsymbol{\kappa}}(\tau)) d\tau$$

Now:

$$\frac{\partial \mathbf{x}}{\partial \kappa_i} = \int_0^t \frac{\partial \mathbf{f}}{\partial \mathbf{u}} \frac{\partial \mathbf{u}}{\partial \kappa_i} + \frac{d\mathbf{f}}{d\mathbf{x}} \frac{\partial \mathbf{x}}{\partial \kappa_i} d\tau$$

where we have omitted the arguments of each function for notational clarity. Taking the time derivative on both sides and noticing that $\frac{\partial \mathbf{x}}{\partial \kappa_i}\big|_{t=0} = \mathbf{0}$ this defines a new dynamical system:

$$\frac{d}{dt} \frac{\partial \mathbf{x}}{\partial \kappa_i} = \frac{d\mathbf{f}}{d\mathbf{x}} \frac{\partial \mathbf{x}}{\partial \kappa_i} + \frac{\partial \mathbf{f}}{\partial \mathbf{u}} \frac{\partial \mathbf{u}}{\partial \kappa_i} \qquad \frac{\partial \mathbf{x}}{\partial \kappa_i}\bigg|_{t=0} = \mathbf{0}$$

note that this is a form of adjoint dynamics. Indeed if we define $\mathbf{a}_i(t) = \frac{\partial \mathbf{x}}{\partial \kappa_i}(t)$ and $\mathbf{h}(\mathbf{x}, t, \kappa) = \frac{\partial \mathbf{f}}{\partial \mathbf{u}} \frac{\partial \mathbf{u}}{\partial \kappa_i}$, then we obtain the final equation:

$$\dot{\mathbf{a}}_i = J_{\mathbf{f}} \mathbf{a}_i + \mathbf{h}(\mathbf{x}, t, \kappa)$$

This is a controlled dynamical system, where the control term $\mathbf{h}$ depends on the solution of the original dynamical system $\mathbf{x}(t)$ and the input $\mathbf{u}(t)$.

$\square$

## S7.4  Some practical considerations

In this section we briefly discuss how this framework can be concretely implemented in an auto-differentiation framework such as Jax or Pytorch. The numerical solution of a dynamical system obtained with, say, an Euler solver, is differentiable with respect to the input to the dynamical system at any time point.[7] In our work, we compute these derivatives in different ways that differ in their computational cost and accuracy:

|  | Adjoint dynamics | Auto-differentiation | Finite difference |
|---|---|---|---|
| Time complexity | $\mathcal{O}(n^2 t)$ | $\mathcal{O}(nt)$ | $\mathcal{O}(nt)$ |
| Memory complexity | $\mathcal{O}(n)$ | $\mathcal{O}(nt)$ | $\mathcal{O}(nt)$ |
| Easy implementation |  | ✓ | ✓ |
| Accurate | ✓ | ✓ |  |

In figure 3 we studied the adjoint analytically, in figures 4 and 6**d**–**h** we used auto-differentiation and in figures 6**c** and S9**a**,**b** finite difference.

## S8 Models

In this section we provide the detailed description of the models used throughout the main manuscript. Jupyter notebooks implementing all analyses performed here are also provided.

### S8.1 Two-layer deep neural network

In section 2 we considered a one-hidden-layer deep neural network given by:

$$y = \tanh(D\mathbf{z}) \in \mathbb{R}^1, \quad \mathbf{z} = \tanh(W\mathbf{x} + \mathbf{b}), \quad \mathbf{x} \in \mathbb{R}^2, W \in \mathbb{R}^{3\times 2}, D \in \mathbb{R}^{1\times 3}$$

The inputs $\mathbf{x}$ are constrained to a circle plus some noise during training:

$$\mathbf{x}(\theta) = [\cos(\theta), \sin(\theta)] + \xi, \quad \theta \in [0, 2\pi), \xi \sim \mathcal{N}(\mathbf{0}, I)$$

where $\mathcal{N}$ is the Gaussian distribution. The loss was given by:

$$\int_0^\pi ||y(\theta) - 1||^2 d\theta + \int_\pi^{2\pi} ||y(\theta) - (-1)||^2 d\theta$$

We evaluated these integrals by discretising $[0, 2\pi)$ into 200 even bins. The parameters were initialised as $w_{i,j}, b_i \sim \mathcal{U}(-1/2, 1/2)$, $d_{i,j} \sim \mathcal{U}(1/3, 1/3)$ where $\mathcal{U}$ is the uniform distribution and optimised using Adam[8] with learning rate 0.01 over 1000 iterations.

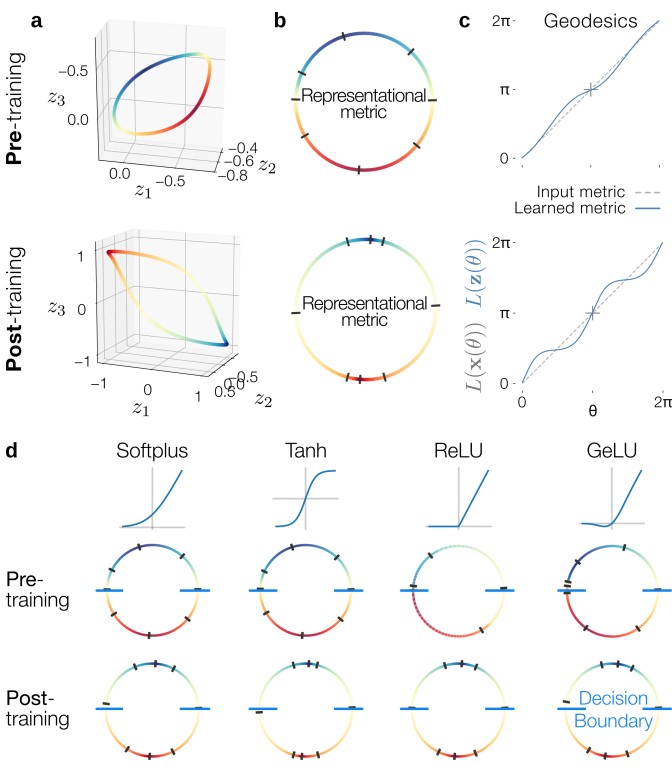

Figure S7: **The representational metric changes over training in deep neural networks.** Top: randomly initialised network. Bottom: network after training. **a.** Activation of the hidden layer in response all inputs. **b.** Learned representational metric visualised a gridlines on the input manifold. **c.** Arclength under either the uniform input metric (dashed line) or learned metric (blue line) integrating from $\theta = 0$ to $\theta = 2\pi$. **d.** Same analysis across different activation functions. Although the intrinsic geometry varies pre-training, the post-training geometry is consistent, hinting at a shared computational mechanism.

### S8.2 E-I network

In section 3 we considered a three-neuron continuous-time RNN following Dales' law, also called an Excitatory-Inhibitory (E-I) network. The dynamics were given by:

$$\dot{\mathbf{x}} = W\phi(\mathbf{x}) - \mathbf{x} + B\mathbf{u}$$

The inputs were constant in time and lied on a 1-d line manifold and the weights were defined according to:[9]

$$W = \begin{bmatrix} a & b & -c \\ b & a & -c \\ d & d & -e \end{bmatrix}, \quad \mathbf{u} = [u, 1-u]$$

with $a, b, c, d \in \mathbb{R}_+$ and $u \in [0, 1]$. We discretized $u \in [0, 1]$ into 200 points and evaluated the corresponding dynamical for each point. To evaluate the 200 dynamical systems we used the Heun second order numerical method[10] until they reached steady-state, saved at 100 time points. This gave a $200 \times 100 \times 3$ mesh that is visualised in panel **c** of figure 3.

### S8.3 Contextual decision-making under noisy inputs

In section 4 we used a classic model of contextual evidence integration.[11] The model is described in the main text. The weights were initialised as $w_{ij} \sim \mathcal{N}(0, g/\sqrt{n})$ the other parameters were initialised as $b_{i,j}, c_{i,j} = \mathcal{N}(0, 1/\sqrt{2})$ and $d_{i,j} \sim \mathcal{N}(0, 1/\sqrt{n})$. We used a differentiable SDE solver with the Heun method and an adaptive step solver.[10] To optimised the system we used Adam in Optax.[12] The hyper-parameters were:

| $n$ | $g$ | $\sigma_{\text{input}}$ | $\sigma_{\text{neuron}}$ | lr | Iterations | Optimiser | SDE solver[10] | PID[10] rtol, atol |
|---|---|---|---|---|---|---|---|---|
| 100 | 0.5 | 0.5 | 0.05 | $5\times10^{-5}$ | 3000 | Adam[8] | Heun | $10^{-2}, 10^{-3}$ |

We next briefly describe the novel analyses performed here.

**The basis of the tangent space.** The basis of the tangent space was constructed using auto-differentiation through the ODE solver. This can be simply achieved in Jax by defining fully differentiable function $\varphi : \mathbb{R}_+ \times \mathcal{M} \to \mathbb{R}^n$ taking points on the manifold of evidences $\mathcal{M}$ and a particular time point in $\mathbb{R}_+$ to return the state of the system at that time point. This function can then be differentiated w.r.t. each of its arguments to obtain the basis of the tangent space at the state of the system at that time point. This function is a composition of i) an embedding of the manifold $\mathcal{M}$ in $\mathbb{R}^2$, which here simply consists of $\mathbf{u}(t) = [u_1, u_2]$ for $u_1, u_2 \in [-0.2, 0.2]$ ii) the evaluation of the ODE under these inputs.

**Variability over noise.** In figure 4**a** (right) and **d** we show the variability of the output of the network to variations to its input. For this we fix a particular Wiener process realisation $\mathcal{W}$ and evaluate the dynamical system by either fixing $u_2 = 0$ and varying $u_1 \in [-0.2, 0.2]$ (for panel **a**) or fixing $u_1 = 0.2$ and varying $u_2 \in [-0.2, 0.2]$ (for panel **d**). The rest of our analyses were performed under a fixed realisation of the noise: the one where $d\mathcal{W}(t) = 0$ for all $t$. Although we note that future work could explore the distribution of the analyses presented here under the measure of this noise process.

**Manifold plotting.** In figure 4**b** and **e** we show slices of the manifold. In **b** they are evidence slices, where $u_1$ and $u_2$ are fixed and we evaluate $\varphi$ varying $t$, in order to obtain a trajectory of the dynamical system. In **e** we take 5 time slices of the manifold by fixing $t$ and evaluating $\varphi$ for all $u_1, u_2 \in [-0.2, 0.2]$. The attractor planes were simply computed by letting $t$ be large enough that the r.h.s. of the dynamical system became close to zero. We note that this is not a "plane attractor" in the sense that the state of the system would tend to those values in the absence of inputs. They are a manifold of attractor fixed points parameterised by $u_1, u_2$.

**The metric.** Under a particular choice of coordinates — here the $[t, u_1, u_2]$ coordinates — the metric at each point on the manifold is a $3 \times 3$ positive (semi-)definite symmetric matrix. In figure 4**e** we plot the metric at two time points on the manifold $t = 0.2$ and $t = 10$ and for $u_1, u_2 = 0$. The results are nevertheless consistent over different values of $u$ (Fig. S8).

**Warping.** In figure 4**f** and **g** we analyse the warping of the manifold across time and conditions. In panel **f** we compute the squared geodesic distance along either the $u_1$ (top) or $u_2$ (bottom) grid lines. For example for the top panels $\int_0^\mu ||\partial_{u_1}\varphi(u_1, u_2)||^2 du_1$ evaluated at $\mu \in \{-0.2, ..., -0.04, 0, 0.04, ..., 0.2\}$. We divided this quantity by the total length of the geodesic to visualise only the effect of warping across values of the input as opposed to the overall stretch or compression of the manifold. Figure S8 shows the same analysis without this division, and integrated from $-0.2$ to $\mu$ (instead of 0 to $\mu$), which further highlights the strength and context dependence of the warping. This is the same analysis performed on the deep neural network of figure 2, except repeated for the different values of the $u$ that is not integrated over. In panel **g** we compute the eigenvalues of the metric. To be more precise, since the metric is a $(0, 2)$-tensor, the usual eigenvalue equation is meaningless and we instead compute the generalised eigenvalues of the metric via the equation $G\mathbf{v}_i = \lambda_i I\mathbf{v}_i$ where is an identity metric. We average them over the irrelevant input, and

then normalised by the top eigenvalue at $u_1 = 0$ in context 1 and $u_2 = 0$ in context 2. The error bars are the eigenvalues over the irrelevant input. The fact that one eigenvalue is much larger at late time points suggests that space is strongly warped into a seemingly 1-dimensional manifold. We note that this effect is less strong away from the decision boundary (Fig. S8), so the warping is not uniform over all relevant input values (as also shown in panel **f**).

**Geometry causally affects computation.** Here, we argue that the warping observed in figure 4 is not simply correlative. To assess whether wrapping was necessary, we trained a network under the constraint that the diagonal elements of the metric remain of equal magnitude — meaning that the representation cannot compress an irrelevant input. Conversely, to assess whether warping is sufficient, we trained a network simply to warp the irrelevant input dimension, before testing the model on the full task. Specifically, this was done by constraining the ratio of the diagonal elements of the metric to be either 1 or their value in the trained RNN. To do that, we defined a new loss $\hat{l}$:

$$\hat{l}_{ctx}(W, A, D) = \alpha l_{ctx}(W, A, D) + \beta \left( c - \frac{G_{u_{ctx} u_{ctx}}}{G_{u_{ctx-1} u_{ctx-1}}} \right)^2$$

where $G_{u_{ctx} u_{ctx}} = \|\partial_{u_{ctx}} x\|^2$ and $u_{ctx}$ is the input of the current context while $u_{1-ctx}$ is the input of the opposite context. When preventing warping, we train all model parameters simultaneously. When we train only on warping, we first train $W$ to warp the representation ($\alpha = 0$, $\beta = 10^{-2}$, $c = 50$) before holding it fixed while training the rest of the model parameters on the original task ($\alpha = 1$, $\beta = 0$).

| Model | | Test performance (MSE, mean $\pm$ std) |
|---|---|---|
| Untrained model | | $0.968 \pm 0.708$ |
| Baseline trained model | $\alpha = 1$, $\beta = 0$ | $0.001 \pm 0.002$ |
| No warping | $\alpha = 1$, $\beta = 10^{-2}$, $c = 1$ | $0.239 \pm 0.257$ |
| Untrained $W$ | | $0.234 \pm 0.259$ |
| Only warping-trained $W$ | $\alpha = 0$, $\beta = 10^{-2}$, $c = 50$ | $0.001 \pm 0.001$ |

Interestingly, warping seems necessary to perform the task, in that the model constrained not to warp does not converge to the optimal task performance. Furthermore, more surprisingly, simply training the recurrent weights of the model to warp the representation, without any task information, is sufficient to reach a performance nearly as good as the baseline model trained on the actual task. Thus, warping is a necessary and sufficient element of the computation of the solution to this task. This further suggests that the warping observed is not purely correlational but indeed the way by which the model performs the task.

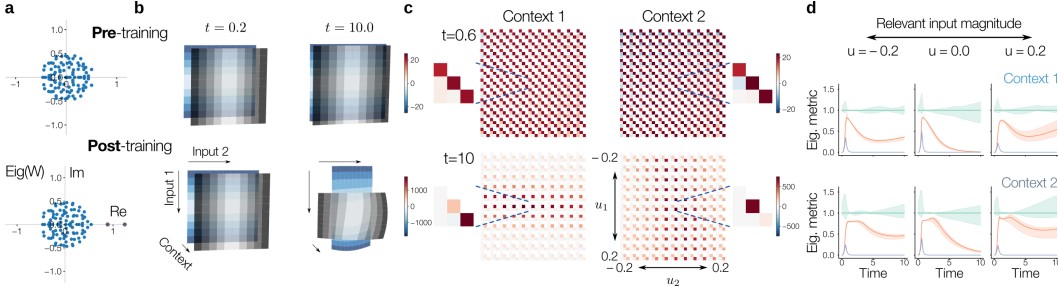

Figure S8: **The context-dependent warping emerges over training and is consistent across inputs. a.** Eigenspectrum of the weights illustrating that the network has few large (real) eigenvalues (circled in red). **b.** Geodesic gridlines from $u_i = -0.2$ in context $i$. The gridlines from the two contexts are overlapping to illustrate that the stretch/compression of the irrelevant input over time is context-dependent. **c.** Metric over all inputs at early and late time points. **d.** Eigenvalues of the metric over different relevant inputs. The error bars are over the irrelevant input.

## S8.4 Sequential working memory

In section 5 we used a common model of working memory[13,14] that is described in the main text. We used the same parameters initialisation as in the contextual evidence integration RNN. The hyper-parameters were:

| $n$ | $g$ | lr | Iterations | Optimiser | ODE solver[10] | PID[10] rtol, atol |
|-----|-----|-----|-----|-----|-----|-----|
| 100 | 0.5 | $5\times10^{-3}$ | 3000 | Adam[8] | Heun | $10^{-2}, 10^{-3}$ |

**Gaussian curvature.** In figure 6c we showed the Gaussian curvature of the torus at a particular time point. In principle the curvature could be computed in a similar way to the metric using adjoint dynamics or automatic differentiation. However such methods would be computationally inefficient, especially to compute the curvature over a fine mesh on the torus. To efficiently compute the Gaussian curvature we used a finite difference method. We evaluated the torus using the ODE solver and fixed step sizes $h = 0.01$ and a mesh of angles $\theta_i \in \{0, \Delta\theta, ...2\pi - \Delta\theta\}$ where $\Delta\theta = 0.01\pi$ (i.e. a mesh size of 200) and computed first order accuracy finite difference estimates of the terms of Gauss's Theorem Egregium formula:

$$\partial_{\theta_1}\mathbf{x} = (\mathbf{x}(t, \theta_1 + \Delta\theta, \theta_2))/\Delta\theta$$
$$\partial_{\theta_2}\mathbf{x} = (\mathbf{x}(t, \theta_1, \theta_2 + \Delta\theta))/\Delta\theta$$
$$V = [\partial_{\theta_1}\mathbf{x}, \partial_{\theta_2}\mathbf{x}] \qquad \text{Basis of tangent space}$$
$$G = V^T V \qquad\qquad \text{Metric}$$
$$\partial_{\theta_1}G = (G(t, \theta_1 + \Delta\theta, \theta_2) - G(t, \theta_1, \theta_2))\Delta\theta$$
$$\partial_{\theta_2}G = (G(t, \theta_1, \theta_2 + \Delta\theta) - G(t, \theta_1, \theta_2))\Delta\theta$$
$$k_1 = (\partial_{\theta_2}G_{\theta_1,\theta_2} - \partial_{\theta_1}G_{\theta_2,\theta_2})/2\sqrt{\det G}$$
$$k_2 = (\partial_{\theta_1}G_{\theta_1,\theta_2} - \partial_{\theta_2}G_{\theta_1,\theta_1})/2\sqrt{\det G}$$
$$\partial_{\theta_1}k_1 = (k(\theta_1 + \Delta\theta, \theta_2) - k(\theta_1, \theta_2))/\Delta\theta$$
$$\partial_{\theta_2}k_2 = (k(\theta_1, \theta_2 + \Delta\theta) - k(\theta_1, \theta_2))/\Delta\theta$$
$$K = (\partial_{\theta_1}k_1 + \partial_{\theta_2}k_2)/\sqrt{\det G} \qquad \text{Gaussian curvature}[15]$$

To illustrate the curvature, we found the points $\theta_1^*, \theta_2^*$ where the curvature was highest (resp. lowest) and took a piece of the torus $\theta_i \in [\theta_i^* - 0.19\pi : \theta_i^* + 0.19\pi]$, defined the matrix $X = [\mathbf{x}(t, \theta_1, \theta_2)]$ for these values of $\theta_1$ and $\theta_2$, and projected $X - \langle X \rangle_{\theta_1,\theta_2}$ on the PC space.

**Plotting the tori.** In figure 6b we plot tori at different time points. They are all color coded according to $\theta_1$. We computed the principal components after mean-centring. That is on $X = [\mathbf{x}(t, \theta_1 = 0, \theta_2 = 0), \mathbf{x}(t, \Delta\theta, 0), \mathbf{x}(t, 0, \Delta\theta), ..., \mathbf{x}(t, 2\pi, 2\pi)]$ where $\Delta\theta = 0.01\pi$, and then projected $X - \langle X \rangle_{\theta_1,\theta_2}$.

**The metric.** In figure 6d we show the metric over the whole torus where the colormap is normalised to the 10th and 90th quantile per entry of the metric. In figure 6e *(top)* we averaged the metric over the torus, while in figure 6e *(bottom)* we first computed $G_{\theta_2\theta_2}/(G_{\theta_1\theta_1} + G_{\theta_2\theta_2})$ before averaging. The error bars showed the 10th and 90th quantile of that value across the points on the torus.

**Subspace similarity.** In figure 6g we computed the similarity between the 2D PC spaces computed at any pairs of time points. The similarity is defined via the nuclear norm $||V(t_1)V(t_2)^T||_*/2$ where $V(t_i)$ is the $2 \times n$ matrix of the two right singular vectors with largest singular values. This defines a notion of average angle between the subspaces.[16]

**Higher-dimensional torus.** In figure 6h,i we trained a new RNN on the same task but now with three targets at three angles $\theta_1, \theta_2, \theta_3$. In figure 6i we show the squared geodesic distance along the gridlines defined by $\theta_i$ for all $i$. They are computed by choosing reference angles (here $(\theta_1, \theta_2, \theta_3) = (\pi, \pi, \pi)$) and computing $L_{\text{right}}^{\theta_i} = \int_\pi^{2\pi} G_{\theta_i\theta_i}(\theta_1, \theta_2, \theta_3)d\theta_i$ (and similarly $L_{\text{left}}^{\theta_i} \int_0^\pi G_{\theta_i\theta_i}(\theta_1, \theta_2, \theta_3)d\theta_i$). The $\theta_i$ side of the prism shown is of length $\langle L_{\text{left}}^{\theta_i} + L_{\text{right}}^{\theta_i} \rangle_{\theta_j,\theta_k}$. The grid shown on the prism is obtained by varying the bound of (say the first) integral and normalising by the total integral $L_{\text{right}}^{\theta_i}$.

**The topology of working memory.** In theorem S7.3 we show that the state of the RNN at a particular time point must lie on the same manifold as the inputs. However, this manifold can be immersed in

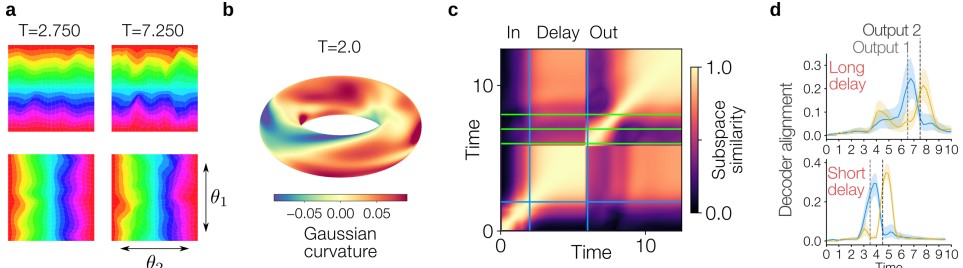

Figure S9: **Working memory dynamically changes representational geometry. a.** Geodesic gridlines at two different time points (mid-delay and end of output) from $\theta_i = \pi$ to $0$ and $2\pi$. The top row are the $\theta_1$ geodesics. Unlike the decision task, there is no stronger warping around a particular point on the manifold, because there is no decision boundary. **b.** Gaussian curvature at the beginning of the delay period. The curvature is more uniform than late in the trial (Fig. 6), but non-zero. **c.** Subspace (2D PC space) similarity during the whole task. **d.** Decoder alignment to the basis of the tangent space under two different delays. It suggests that it is the go signal that cues this alignment.

the neural state space (e.g. intersecting itself), or quotiented (e.g. if there is periodicity). Here we show that it must be exactly a torus at any time point after the second input presentation, and that is embedded in the state space. For this it suffices to prove that $\mathbf{x}(t, \theta_1, \theta_2)$ is a homeomorphism in its arguments.

**Proposition S8.1.** *Let $\dot{\mathbf{x}}$ a working memory RNN defined as in section 5, and suppose the RNN performs the task correctly, meaning that $D\phi(\mathbf{x}(t_1, \theta_1, \theta_2)) = [\cos(\theta_1), \sin(\theta_1)]$ where $t_1$ is the first output time (and similarly for the second output). Then throughout the delay and output period the state of the RNN lies on a (hyper-)torus embedded in the RNN states-space $\mathbb{R}^n$ at any time point.*

We will prove this for a 2-torus (i.e. 2 targets), the proof generalises naturally to an arbitrary number of targets.

*Proof.* After the second input has been presented the system is a composition of autonomous flows: $(\varphi_{t_3}^3 \circ \varphi_{t_2}^2 \circ \varphi_{t_1}^1)(\mathbf{x}(t_0, \theta_1, \theta_2))$ where $t_0$ is the time point where the second input is removed, $t_1$ is the delay, $t_2$ the duration of the go cue and $t_3$ the time until the end of the trial. Each flow is autonomous and therefore a time- and initial-state diffeomorphism by lemma S7.4. Furthermore, a composition of diffeomorphic flows is diffeomorphic — and in particular homeomorphic. Therefore, the manifold will be an embedded torus if it is an embedded torus at a particular time point.

In theorem S7.3 we have shown that $\mathbf{x}(t, \theta_1, \theta_2) = P(\mathbf{y}(t, \theta_1, \theta_2))$ where $P$ is a projection map and $\mathbf{y}(t, \theta_1, \theta_2)$ lies on a manifold of the same topology as its input, here a torus. It therefore remains to prove that $P$ is a bijection from the torus to its image (i.e. injective).

Let $(\theta_1^1, \theta_2^1), (\theta_1^2, \theta_2^2)$ such that $\mathbf{x}(t, \theta_1^1, \theta_2^1) = \mathbf{x}(t, \theta_1^2, \theta_2^2)$. Then, by time-diffeomorphism, $\mathbf{x}(t_1, \theta_1^1, \theta_2^1) = \mathbf{x}(t_1, \theta_1^2, \theta_2^2)$ where $t_1$ is the first output time. We can now apply the decoder on each side: $D\phi(\mathbf{x}(t_1, \theta_1^1, \theta_2^1)) = D\phi(\mathbf{x}(t_1, \theta_1^2, \theta_2^2))$. By assumption, the RNN performs the task correctly, so this equality holds if and only if $\theta_1^1 = \theta_1^2$. By the same argument for the time of the second output $\theta_2^1 = \theta_2^2$. Thus by the time diffeomorphism, for all $t$, $\mathbf{x}(t, \theta_1^1, \theta_2^1) = \mathbf{x}(t, \theta_1^2, \theta_2^2)$ if and only if $(\theta_1^1, \theta_2^1) = (\theta_1^2, \theta_2^2)$. This shows that $\mathbf{x}$ is a topological embedding.

$\square$

## S8.5 Parametric working memory

We trained a model on a parametric working memory task called the Romo task.[17,18] The RNN sequentially receives inputs of varying magnitude through the same input vector.

$$\dot{\mathbf{x}} = W\phi(\mathbf{x}) - \mathbf{x} + \mathbf{b}u_1 \mathbb{1}_{0 \leq t \leq 0.5} + \mathbf{b}u_2 \mathbb{1}_{1.5 \leq t \leq 2.0}$$

where $u_1, u_2 \in [0.5, 1.0]$. The task of the network is to output $0$ until the end of a delay period, and then the sign of the difference between $u_1$ and $u_2$ afterwards, as encoded via a one-hot vector (Fig. S10**a**).

$$L(W, A, D) = \int_0^{T_1} ||D\phi(\mathbf{x}(t))||^2 dt + \int_{T_2}^{T_3} ||D\phi(\mathbf{x}(t)) - [\mathbb{1}_{u_1 > u_2}, \mathbb{1}_{u_1 < u_2}]||^2 dt$$

where $T_1 = 4.0$ is the end of the delay, $T_2 = 4.5$ and $T_3 = 7.0$ the end of the trial. Here $D \in \mathbb{R}^{n \times 2}$ and $\phi$ is the tanh nonlinearity.

| $n$ | $g$ | lr | Iterations | Optimiser | ODE solver[10] | dt |
|-----|-----|-----|-----|-----|-----|-----|
| 100 | 0.5 | $10^{-3}$ | 2000 | Adam[8] | Heun | $10^{-2}$ |

We found that the dimensionality of the manifold expanded during the delay (Fig. S10**b**, bottom). Indeed, the metric became a multiple of the identity matrix (Fig. S10**b**, top), suggesting that the input which was originally encoded along a single vector **b** had now been stored along two locally orthogonal directions $\partial_{u_1}\mathbf{x}$ and $\partial_{u_2}\mathbf{x}$ of similar magnitudes, on the manifold. However, by the time of the output period the manifold had collapsed to become intrinsically 1-dimensional (Fig. S10**b**, *bottom*). Furthermore, the partial derivatives $\partial_{u_1}\mathbf{x}$ and $\partial_{u_2}\mathbf{x}$ pointed in opposite directions (Fig. S10**b**, *top*). This is consistent with the network collapsing the representation of the two inputs onto a line representing the output. We can consider the pullback of the subtraction operation: suppose that the two sequential inputs have been mapped to orthogonal vectors $\mathbf{v}_1, \mathbf{v}_2 \in \mathbb{R}^n$ such that the neural representation is given by $\mathbf{x}(u_1, u_2) = u_1\mathbf{v}_1 + u_2\mathbf{v}_2 \in \mathbb{R}^n$ and consider the subtraction operation $\varphi : \mathbb{R}^2 \to \mathbb{R}^n$ given by $\varphi(u_1, u_2) = u_2\mathbf{v}_2 - u_1\mathbf{v}_1$. Then the pullback metric is:

$$G(u_1, u_2) = J_\varphi^T J_\varphi = [1, -1] \otimes [1, -1] = \begin{bmatrix} 1 & -1 \\ -1 & 1 \end{bmatrix}$$

which matches the form of metric of the manifold near the output time (Fig. S10**b**, *top*). The sign operation and the mapping to the one-hot encoded outputs can be performed via the decoder $D\phi(\cdot)$, consistent with previously reported partitions of task computations into linear and binarisation operations.[19]

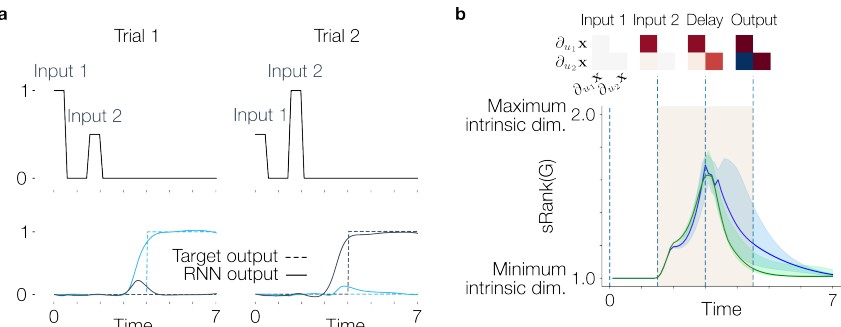

Figure S10: **Neural dimensionality expands to store working memories before collapsing to task outputs. a.** *Romo task*: the network receives two inputs sequentially and then has to output the one-hot encoded sign of the larger input. **b.** Intrinsic dimensionality of the manifold, as quantified by the stable rank (sRank) of the metric, that is $\frac{\|G\|_*}{\|G\|_2}$. Blue is over the submanifold where the output is $[0, 1]$, i.e. $u_2 > u_1$. Green is over the submanifold where the target output is $[1, 0]$, meaning $u_1 > u_2$. The error bars are the 0.2 and 0.8 quantiles.

### S8.6 State-space model on brain-computer interface data

To illustrate the usefulness of our method on data applications we trained a state-space model (SSM) on brain-computer interface data. The data were recorded from a human subject instructed to draw centre-out lines.[20] We found that a one-layer HiPPO (LegS) SSM[21] of the form:

$$\dot{\mathbf{x}} = A\mathbf{x} + B\mathbf{u}(t), \qquad L(A, B) = \sum_{i=1}^{T} \|D\mathbf{x}(t_i) - \mathbf{y}(t_i)\|^2$$

Where $\mathbf{x}(t)$ is the state of the SSM and $\mathbf{u}(t)$ is the time-interpolated input and $\mathbf{y}(t_i)$ the target cursor position. Since the task is performed offline, we defined the target cursor position as $\mathbf{y}(t) =$

$r(t, l)[\cos\theta, \sin\theta]$ where the radius $r$ is given by:

$$r(t) = \begin{cases} 0 & t < t_{\text{onset}} \\ \frac{l}{1+e^{-(20t-6)}} & t_{\text{onset}} \leq t \leq t_{\text{offset}} \\ l & t > t_{\text{offset}} \end{cases}$$

where $l \in [1, 3]$ is the length of the line.

| $n$ | lr | Iterations | Optimiser | ODE solver[10] | dt |
|-----|-----|-----------|-----------|----------------|-----|
| 100 | $10^{-3}$ | 2000 | Adam[8] | Heun | $10^{-2}$ |

For a fixed line length, our theory predicts that after the movement onset the state of the SSM will lie on a manifold of cylindrical topology representing the time $t$ and angular $\theta$ variables (Fig. S11**a**). To study how different line lengths affect the dynamics, we interpolated the neural data using Akima cubic splines[22] to define a continuous-in-line-length input $\mathbf{u}_l(t)$ to the SSM. Computing the time-length metric reveals that, during movement, the time and length partial derivatives become strongly aligned, suggesting that to perform longer lines neural activity accelerates (Fig. S11**b**). Such accelerating neural trajectories have been previously observed in RNNs and neural data from timing tasks.[23] We thus illustrate that computing the metric allows to directly probe for this effect.

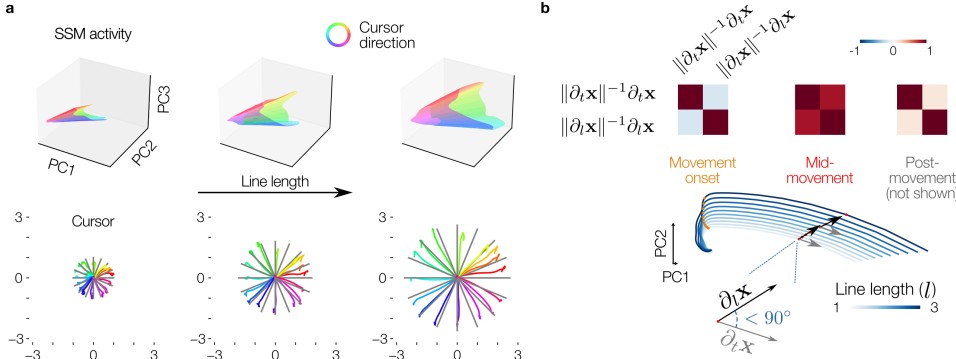

Figure S11: **SSM trajectory speed varies to generate BCI cursor movements of different lengths. a.** *Top:* SSM activity in PCA space for a short timespan after movement onset. *Bottom:* Decoded cursor position. **b.** Length-time metric. At movement onset the time $t$ and line length $l$ variables are locally encoded in orthogonal directions, while during movement they are strongly aligned.

## Supplementary References

1. Lee, J. M. and Lee, J. M., Smooth manifolds, Springer, 2003.

2. Hauser, M. and Ray, A. 2017, Principles of Riemannian Geometry in Neural Networks, Advances in Neural Information Processing Systems, ed. by Guyon, I. et al., vol. 30.

3. Zangrando, E. et al. 2024, Neural rank collapse: Weight decay and small within-class variability yield low-rank bias, arXiv preprint arXiv:2402.03991.

4. Aubry, M. and Russell, B. C. 2015, Understanding deep features with computer-generated imagery, Proceedings of the IEEE international conference on computer vision, pp. 2875–2883.

5. Hentschel, H. G. E. and Procaccia, I. 1983, The infinite number of generalized dimensions of fractals and strange attractors, Physica D: Nonlinear Phenomena **8**(3), 435–444.

6. Kunita, H. and Ghosh, M., Lectures on stochastic flows and applications, vol. 78, Tata Institute of Fundamental Research Bombay, 1986.

7. Chen, R. T. et al. 2018, Neural ordinary differential equations, Advances in neural information processing systems **31**.

8. Kingma, D. P. 2014, Adam: A method for stochastic optimization, arXiv preprint arXiv:1412.6980.

9. Wong, K.-F. and Wang, X.-J. 2006, A recurrent network mechanism of time integration in perceptual decisions, Journal of Neuroscience **26**(4), 1314–1328.

10. Kidger, P. 2022, On neural differential equations, arXiv preprint arXiv:2202.02435.

11. Mante, V. et al. 2013, Context-dependent computation by recurrent dynamics in prefrontal cortex, nature **503**(7474), 78–84.

12. DeepMind, The DeepMind JAX Ecosystem, 2020.

13. Stroud, J. P. et al. 2023, Optimal information loading into working memory explains dynamic coding in the prefrontal cortex, Proceedings of the National Academy of Sciences **120**(48), e2307991120.

14. Cueva, C. J. et al. 2021, Recurrent neural network models for working memory of continuous variables: activity manifolds, connectivity patterns, and dynamic codes, arXiv preprint arXiv:2111.01275.

15. Pressley, A. N., Elementary differential geometry, Springer Science & Business Media, 2010.

16. Hitzer, E. 2013, Angles between subspaces, arXiv preprint arXiv:1306.1629.

17. Romo, R. et al. 1999, Neuronal correlates of parametric working memory in the prefrontal cortex, Nature **399**(6735), 470–473.

18. Schuessler, F. et al. 2020, The interplay between randomness and structure during learning in RNNs, Advances in neural information processing systems **33**, 13352–13362.

19. Brandon, J., Chadwick, A., and Pellegrino, A. 2025, Emergent Riemannian geometry over learning discrete computations on continuous manifolds, arXiv preprint arXiv:2512.00196.

20. Willett, F. R. et al. 2021, High-performance brain-to-text communication via handwriting, Nature **593**(7858), 249–254.

21. Gu, A. et al. 2022, How to train your hippo: State space models with generalized orthogonal basis projections, arXiv preprint arXiv:2206.12037.

22. Akima, H. 1970, A new method of interpolation and smooth curve fitting based on local procedures, Journal of the ACM (JACM) **17**(4), 589–602.

23. Wang, J. et al. 2018, Flexible timing by temporal scaling of cortical responses, Nature neuroscience **21**(1), 102–110.

