# OpenReview forum: "RNNs perform task computations by dynamically warping neural representations"
_NeurIPS.cc/2025/Conference — NeurIPS 2025 poster_

### Official Review · Reviewer_2FTC · 2025-06-11

**Clarity:** 4
**Significance:** 2
**Originality:** 3
**Rating:** 4
**Confidence:** 3

**Summary:**

The paper introduces a framework for understanding computation in recurrent neural networks (RNNs) based on Riemannian geometry. The central hypothesis is that RNNs perform computations by dynamically warping the geometry of their internal neural representations. The authors develop a mathematical formalism to test this hypothesis. First, they show that if the time-varying inputs to a dynamical system lie on an $m$-dimensional manifold, the trajectories of the state are constrained to a manifold of at most $m+1$ dimensions. Then, they define a "representational metric" on this neural activity manifold, derived as a pullback from the state space. This metric quantifies the local "warping" of the representation space and can be computed by solving an adjoint differential equation alongside the main RNN dynamics.

The framework is applied to two toy tasks: In the first task, the model has to output the sign of one of two real inputs, selected based on a (one-hot) contextual clue. The analysis reveals that the network learns to warp its representational manifold to selectively compress the dimension corresponding to the irrelevant input, while expanding the dimension for the relevant input near the decision boundary (close to zero). In the second task, the model has to reproduce two inputs after a time delay. The authors show that the network activity lies on a toroidal manifold, which dynamically changes its intrinsic geometry to selectively encode and recall items from memory.

**Questions:**

1. Have you considered applying your framework to RNNs trained on real-world tasks, for example, for natural language processing? I think it could be very insightful to explore how different architectures (e.g., LSTMs vs. SSMs) solve tasks. Maybe it would even be possible to extend the framework, so that it can be applied to (autoregressive) transformers? They can be considered as a special kind of RNN, where the size of the internal state (the KV-cache) grows over time (this paper https://arxiv.org/abs/2006.16236 is helpful to make this analogy clearer).
2. Do you see a path toward using these geometric insights prescriptively? For instance, could the representational metric be incorporated into the training process (perhaps as a sort of regularizer), to encourage a model to learn specific geometric structures and potentially improve learning speed and/or generalization?
3. Under Limitations, you mention the degeneracy of the map from network parameters to geometry. Have you investigated whether different parameter initializations converge to solutions with similar geometric properties? Perhaps more complicated tasks are necessary to observe "distinct solutions" (for the investigated toy tasks, it is difficult to think of different solutions).

Typos (these do not influence the rating):

- line 231: "context-depenent" should be "context-dependent"
- caption of Figure 6b: "is lies" should be "lies"

**Ethical Concerns:**

["NO or VERY MINOR ethics concerns only"]

**Final Justification:**

The authors addressed many of the weaknesses I raised in my original review. I still believe if the authors looked at "more relevant" tasks (e.g., language modelling), it would have made the paper potentially more interesting for a wider audience, but I respect the author's decision to stick to the neuroscience theme of the paper. I think the paper could be accepted, but will only be of interest for a relatively narrow audience (hence the borderline rating).

**Limitations:**

yes

**Quality:**

2

**Strengths And Weaknesses:**

**Strengths**
1. The paper is well-written and organized. Many helpful illustrations allow the reader to develop an intuitive understanding of the relevant concepts and ideas.
2. The mathematical framework is rigorously developed and intuitively makes sense. It goes beyond an analysis of the dynamis near fixed points.

**Weaknesses**
1. The authors only apply their framework to very small models and simple toy tasks. While this is helpful to illustrate central concepts, an application to larger models trained to solve real-world tasks is lacking. Without this, it remains unclear whether the framework can be used to gain an understanding of the inner workings of more complex models.
2. While the framework provides an account of how a model's internal representations are transformed, the link between geometry and computation seems correlational: While the geometry warps *as* the model performs the task, it is not evident to me that this geometric transformation *is* the computation itself, versus a by-product of another underlying process (i.e. the "actual computation"). The presented framework is currently post-hoc, and it is not yet clear if it can be used prescriptively to design or train models more effectively.

While I find the general premise of the paper very interesting, my main concern (and why I did not rate it higher) is that it feels "incomplete" to me. The paper does not apply the developed framework to any "real problem" and does not demonstrate what kind of improvements or advancements it enables. I think the paper would be much more impactful if it demonstrated a few applications beyond toy examples.

---

> ### Author Rebuttal · Authors · 2025-07-31
>
> **Summary.** We thank the reviewer for his supportive comments and the overall feedback he provided on our work. We are particularly pleased that he found that “the mathematical framework is rigorously developed” going “beyond an analysis of the dynamics near fixed points”, while still "developing an intuitive understanding.”
>
> The main questions raised by the reviewer were:
>
> * Whether our framework could be applied to more complex models and tasks
> * Whether studying the geometry of neural networks could provide insights into computational mechanisms, beyond correlations
>
> To  address these comments we have now:
>
> * Added an application of our framework to state-space models (SSMs) trained on a brain-computer interface (BCI) task.
> * Added an analysis where we explicitly manipulate the geometry of RNN activity to show that warping can be a necessary and sufficient component of task computation
>
> We thank the review for these comments as we believe that these additional analyses have helped make the work more appealing to a broader audience, and that it now better highlights warping as a fundamental component of task computation in dynamical systems.
>
> ___
> **Applications to more complex models.** We thank the reviewer for suggesting additional applications, and confirm that our Riemannian geometry framework is applicable to a large class of dynamical systems and tasks. We agree with the reviewer that an example network trained on real-world data would make the work more appealing to a broader audience. We have therefore added a **new application of our framework to state-space models trained on a brain-computer interface task**.
>
> The SSM is trained to map high-dimensional time series data of brain activity to intended cursor velocities on a screen. We adapted an existing SSM architecture (POSSM \[1\]), to this task. We chose this task to provide an application to real-world data while still remaining within the computational neuroscience theme of the paper.
>
> Because speeds and directions of the cursor are central to BCI tasks, we asked how the SSM model represented them. For this, after training the model, we looked at a subset of the time-series data, where the human subject systematically varied the intended heading angle and velocity of the cursor, starting at the center of the screen. Thus, the neural signals should approximately lie on the manifold $\\mathcal{M}=S^1 \\times \\mathbb{R}$, representing the direction and speeds of the cursor. This corresponds to a slice of the full manifold of all possible cursor movements.
>
> We derived the representational metric of this manifold by deriving the pullback from the SSM state-space to $\\mathcal{M}$. Doing so revealed that different velocities correspond to different warpings of the neural manifold representing these velocities. We found that the non-diagonality was high, which suggested that the encoding of speed and time on the manifold were strongly correlated. In other words, to make cursor movements faster, the neural activity accelerates. This phenomenon had been previously reported in the neuroscience literature in other timing tasks \[2\]. This corresponds to a warping of the neural manifold along the time dimension as the speed increases.
>
> Thus, our framework can be applied to more complex models trained on larger tasks. The key step is to define the manifold that one wishes to study — here velocities and directions. In general this manifold will be a submanifold of a larger manifold of inputs over which the model was trained. For example, previous work has tried understanding vision models trained on large-scale image classification, and where the manifold of interest was the position or orientation of the object in the image \[3\]. Furthermore, LLMs where the manifold of interest might be a constrained set of inputs for a fixed prompt \[4\].
>
> The paper mentioned by the reviewer linking transformers and RNNs — which we have now cited —  suggests that the tools we have developed could in principle be applied to those models.
>
> **“Toy-tasks”.** While we believe that the additional application to real world data suggested by the reviewer significantly improves our work, we note that the “toy-tasks” considered in this paper are well-studied and fundamentally important problems in neuroscience. For example, the contextual decision-making tasks we considered are used in the computational neuroscience \[5\] and machine learning \[6-7\] literature to probe the inner-workings of biological and artificial neural networks. These tasks, although simple, are not trivial. They require particular computations which the neural network — whether biological or artificial — must implement through specific patterns of neural activation.
>
> To better highlight that the models presented, although simple, are tackling “real problems”, we have now discussed in the introductory paragraph to each section how the model and task we analyse have been previously used in the computational neuroscience and machine learning literature to understand dynamical neural computations.
>
> ___
> **Insights beyond correlations.** We agree with the reviewer that our study is correlational: we observe that in multiple models, the representation is warped as the  task is performed, and that this warping correlates with the computations performed in the task. For example, in the contextual decision-making task, we found that the manifold is warped (compressed) along the context-irrelevant input. However this warping could in principle be a by-product of another more fundamental mechanism by which the model performs the task.
>
> To further highlight that **warping is not an epiphenomenon** but the fundamental mechanisms by which the network performs the task, we’ve now performed the following analyses in the contextual decision-making RNN:
>
> * To assess whether wrapping was necessary, we’ve now trained a network under the constraint that the diagonal elements of the metric remain of equal magnitude — meaning that the representation cannot compress an irrelevant input.
> * To assess whether warping is sufficient, we’ve now trained a network simply to warp the irrelevant input dimension, before testing the model on the full task.
>
> We’ve done this analysis by constraining the ratio of the diagonal elements of the metric to be either 1 or their value in the trained RNN. To do that, we’ve added a term to the loss: $\lVert c - G_{u_{ctx}u_{ctx}}/G_{u_{ctx-1}u_{ctx-1}}\rVert$ where $G_{u_{ctx}u_{ctx}}=\lVert \partial_{u_{ctx}} x\rVert^2$ and $u_{ctx}$ is the input of the current context while $u_{ctx-1}$ is the input of the opposite context.
>
> |  | Untrained model | Baseline trained model | No warping | Untrained $W$ | Only warping-trained $W$ |
> | :---- | :---- | :---- | :---- | :---- | :---- |
> | Test performance (MSE, mean $\\pm$ std) | $0.968 \\pm 0.708$ | $0.001 \\pm 0.002$ | $0.239 \\pm 0.257$ | $0.234 \\pm 0.259$ | $0.001 \\pm 0.001$ |
>
> Interestingly, warping seems necessary to perform the task, in that the model constrained not to warp does not converge to the optimal task performance. Furthermore, more surprisingly, simply training the model on warping is sufficient to reach a performance nearly as good as the baseline model trained on the actual task. Thus, **warping is a necessary and sufficient element of the computation** of the solution to this task. This further suggests that the warping observed is not purely correlational but indeed the way by which the model performs the task.
>
>
> ___
>
> **Robustness of the metric across initialisations and model hyperparameters.** We have now added an analysis showing the robustness of the metric to different random seeds. The geodesic distance between points on the manifold is very similar across model initialisations. In the network from section 2, we find an expected geodesic distance difference of:  $E\\left\[\\int\_0^{2\\pi} \|\\gamma\_1(\\theta)-\\gamma\_2(\\theta) \|d\\theta\\right\]=0.0227$.
>
> We can nevertheless get distinct solutions by varying the network hyperparameters. For example, previous work has suggested that varying the magnitude of the initialisation can make the network transition between a rich learning regime where it learns a general solution to the task to a lazy regime where it overfits particular inputs \[8\]. We find that the intrinsic geometries under different initialisations are significantly different $E\\left\[\\int\_0^{2\\pi}\| \\gamma\_1(\\theta)-\\gamma\_2(\\theta) \|d\\theta\\right\]=0.19498597$, close to the difference between a trained and random network $E\\left\[\\int\_0^{2\\pi} \|\\gamma\_1(\\theta)-\\gamma\_2(\\theta) \| d\\theta\\right\]=0.273$.
>
> Thus, while the metric is generally robust across seeds, one can design models with qualitatively different geometries (corresponding to different generalisation abilities) by varying model hyperparameters.
>
> ___
>
> **References**
>
> \[1\] Ryoo, A. H. et al. (2025). Generalizable, real-time neural decoding with hybrid state-space models.*NeurIPS*
>
> \[2\] Beiran, M.et al (2023). Parametric control of flexible timing through low-dimensional neural manifolds. *Neuron*.
>
> \[3\] Kaul, P., & Lall, B. (2019). Riemannian curvature of deep neural networks. *IEEE*.
>
> \[4\] Li, K. et al. (2023). Emergent world representations: Exploring a sequence model trained on a synthetic task. *ICLR*.
>
> \[5\] Mante, V. et al. (2013). Context-dependent computation by recurrent dynamics in prefrontal cortex. *Nature*.
>
> \[6\] Valente, A. et al. (2022). Extracting computational mechanisms from neural data using low-rank RNNs. *NeurIPS*.
>
> \[7\] McMahan, B. et al. (2021). Learning rule influences recurrent network representations but not attractor structure in decision-making tasks. *NeurIPS*.
>
> \[8\] Saxe, A. et al. (2013). Exact solutions to the nonlinear dynamics of learning in deep linear neural networks. *arXiv*.

---

> > ### Comment · Reviewer_2FTC · 2025-08-01
> >
> > I thank the authors for the additional experiments they performed and the clarifications they will add to their manuscript. I think these changes greatly improve the work.

---

> > > ### Author Response · Authors · 2025-08-05
> > >
> > > We are glad that we addressed the reviewer's comments and we thank them for their positive feedback.

---

### Official Review · Reviewer_aFgg · 2025-06-26

**Clarity:** 4
**Significance:** 3
**Originality:** 3
**Rating:** 5
**Confidence:** 3

**Summary:**

The authors present a formal framework for understanding computation through dynamics present in recurrent neural network architectures with time-varying inputs. They study the dimensionality and topology of network dynamics with respect to the structure of the input data for both contextual decision making and working memory tasks. On simple models of a contextual classification and memory task, they demonstrate how the geometry of the dynamics is warped to allow the network to discard (contextually defined) irrelevant information through dynamics.

**Questions:**

- In section 2, does the choice of Tanh non-linearity have a significant effect on the warping of the space near the decision boundary compared to a ReLU? More generally, can the authors comment on how the choice of piece-wise linear non-linearity will impact their results compared with 'sigmoid-like' non-linearities?
- It appears a proof of Theorem 3.4 is lacking from the supplementary material. Is this somehow given by 3.3? Should it then be written as a corollary?
- Can the authors comment on the relationship of their theory to that of Low-Rank RNNS? Adding a discussion of this to the paper would be beneficial for the community.

**Ethical Concerns:**

["NO or VERY MINOR ethics concerns only"]

**Final Justification:**

The paper is well written, technically sound, creative, and a novel method for understanding computation in RNNs. The authors have addressed all my concerns regarding the relation to prior work with exceptional clarity, and have provided additional experiments. I believe the paper is a clear benefit to the community and would make a strong contribution to the conference.

**Limitations:**

Yes

**Paper Formatting Concerns:**

None.

**Quality:**

4

**Strengths And Weaknesses:**

**Strengths:**
- The formal study of time-varying inputs in RNN dynamical systems is indeed something which is lacking in the literature, and a valuable contribution, especially to the computational neuroscience community (which the authors appear well aware of).
- The framework seems broadly applicable to the study of nonlinear dynamics, and therefore has the potential to yield novel insights not possible with current methods.
- The paper is well written and organized for readability and understanding. For example, the review of core concepts for static neural representations in Section 2 is a welcome addition.
- The proposed framework and viewpoint for understanding computation in recurrent neural networks is novel to the best of my knowledge and appears to open many doors for further understanding how RNNs perform computation -- the authors also clearly demonstrate this potential through their chosen examples.


**Weaknesses**
- The paper seems to have a relatively significant connection to the recent work on Low-Rank RNNs from Valente, Pillow & Ostojic, (Neurips, 2022). The authors cite this paper, yet have no further discussion on the relationship to that theory.

---

> ### Author Rebuttal · Authors · 2025-07-31
>
> **Summary.** We thank the reviewer for his positive comments. We are particularly pleased that he found our work to be “broadly applicable to the study of nonlinear dynamics” such that it has “the potential to yield novel insights not possible with current methods”, and that he generally found it to be “well written.”
>
> The reviewer raised some questions regarding:
>
> * The relationship between our work and the recent work on low-rank RNNs (e.g. Valente, Pillow, Ostojic, 2022, *NeurIPS*)
> * The effect of the choice of nonlinearity on the geometry (especially in the model from section 2\)
>
> We’ve now addressed these questions by:
>
> * Improving our discussion and background sections to better relate our work to the low-rank RNN framework
> * Providing additional simulations showing how various nonlinearities affect the intrinsic geometry.
>
> We thank the reviewer for very positively receiving our work, and we believe that the revisions stemming from his comments have helped further highlight the novelty and generality of our work.
>
> We have provided below point-by-point responses to the reviewers questions.
>
> ___
> **Relationship to the low-rank RNN framework.** We thank the reviewer for raising this point, and agree that a deeper discussion of the relationship to work on low-rank RNNs will be of substantial benefit to readers in the computational neuroscience community. The key distinction between our work and low-rank RNNs is that low-rank RNNs restrict the *embedding* dimensionality of RNN activity, while our work mathematically characterises the *intrinsic* dimensionality.
>
> Indeed, if the connectivity of an RNN is low rank, meaning that $W=\\sum_{i=1}^r \\mathbf{u}_i\\mathbf{v}_i^T$, then the dynamics of the RNN upon receiving a d-dimensional input $\\mathbf{a}(t)$ can be written as
>
> $$\\frac{d\mathbf{x}}{dt}=-\mathbf{x}+W\\phi(\mathbf{x})+B\\mathbf{a}(t)=-\mathbf{x}+\\sum_{i=1}^r \\mathbf{u}_i (\\phi(\\mathbf{x})\\cdot \\mathbf{v}_i) + \\sum_i^d \\mathbf{b}_i a_i(t)$$
>
> which is a linear combination of the $\\mathbf{u}$'s and $\\mathbf{b}$'s (plus a leak term). In particular, this means that the dynamics of low-rank RNNs are constrained to an $r+d$ dimensional subspace, with activity in all other dimensions rapidly decaying due to leak. Furthermore, this means that if the system is initialised within this subspace, the state of the low-rank RNN remains embedded in this $r+d$-dimensional subspace.
>
> Crucially, in low-rank RNNs the input $\mathbf{a}(t)$ is $d$-dimensional at any particular time point. Instead, we constrain the inputs to the system to lie on an $m$-dimensional manifold of time-varying functions which can take on arbitrary values in n-dimensional state space. That is, each point on the manifold is a different function. In table 1 of our manuscript we show the $d$ and $m$ dimensionality of various models and tasks typically considered in computational neuroscience. In general, $d$ can be high while $m$ is low, and vice versa. In addition, we show that activity manifolds are low-dimensional even when connectivity is full-rank and inputs are high-dimensional in state space, provided that the inputs are constrained to a low-dimensional manifold of functions.
>
> The two frameworks can be naturally united if we combine their assumptions, that is if:
>
> 1. The connectivity matrix W is low-rank — of rank r
> 2. The instantaneous input $\mathbf{a}(t)$ lies on a low-dimensional space — of dimension $d$
> 3. The input function $\\mathbf{a}$ lies on a low-dimensional manifold (in function space) — of dimension $m$.
>
> In this case, by the results we derive (Theorem 3.1, S7.3, and corollary S7.5), the state of the system lies on a low-dimensional nonlinear manifold of dimension $m+1$, and via the theorems derived in Ostojic’s work \[1-2\] this manifold will be embedded within an $(r+d)$-dimensional subspace of the RNN state-space. In other words, the low-rank RNN framework places constraints on the linear embedding dimensionality of the state while our framework places constraints on the intrinsic dimensionality of the nonlinear manifold of states the RNN can take.
>
> We believe that future work could take advantage of the natural link between these frameworks to design models that better match experimental data. For example, one may find — perhaps using dimensionality reduction methods — that recordings from a particular brain region during a particular task are embedded in a $d$-dimensional space, but also lie on a $m$-dimensional nonlinear manifold. Then, one could design an RNN model that matches these experimentally observed phenomena. This could be made possible by the extensive line of work characterising the intrinsic and embedding dimensionality of neural activity \[3-4\]. Future works could also attempt to understand how placing constraints on the embedding dimensionality by imposing low-rank connectivity affects the intrinsic geometry of the nonlinear manifold.
>
> *We think that this connection between the well-established low-rank RNN framework and our framework is a strength, which could help the reader gain intuition about how the manifolds we discussed are defined. We have therefore expanded on this link in the introduction and background sections, including additional references to the work we cite below, and a new schematic introducing the links between intrinsic and embedding dimensionality.*
>
> ___
> **The effect of the choice of nonlinearity in section 2\.** We agree with the reviewer that it is important to understand how model choices affect the geometry. We have now added a supplementary material figure where we re-train the model from section 2 with various nonlinearities (ReLU, Tanh, Softplus and GeLU).
>
> We found that the intrinsic geometry of the manifold was very stable across nonlinearities. We’ve included below a table showing the difference in geodesic distance (which is a quantity between 0 and pi/2) between the baseline tanh model and these other models.
>
> |  | Tanh | Tanh (different seed) | Softplus | ReLU | GeLU | Tanh (pre-training) |
> | :---- | :---- | :---- | :---- | :---- | :---- | :---- |
> | Difference in geodesic distances from decision boundary $\int_0^\pi \|\gamma_{\text{model 1}}(\theta)-\gamma_{\text{model 2}}(\theta)\| d\theta$ | 0 | $0.007 \\pm 0.007$ | $0.031 \\pm 0.042$ | $0.009 \\pm 0.013$ | $0.022 \\pm 0.031$ | $0.273 \\pm 0.229$ |
>
> We interpret this as meaning that the warping is closely tied to the computation performed by the network: small changes in input angles near the decision boundary will correspond to large changes in the hidden layer activity, in turn corresponding to switching from an output of \-1 to 1\. Still, in more complex tasks, we expect that different networks could use fundamentally different solutions, which have different intrinsic geometries.
>
> ___
> **Proof of theorem 3.4.** We thank the reviewer for catching this. The proof of theorem 3.4 directly follows from that of theorem 3.3 with one additional step of derivation. To make that clearer, we have now added this step as a separate proof in supplementary material, and as suggested by the reviewer, we have now relabelled this result as a corollary.
>
> **Overall.** We thank the reviewer for his positive and constructive comments which have helped improve the work. In particular, we believe that the discussion of the relationship between our work and the low-rank RNN framework will help explain our framework to a broader audience, and that the additional validations of the effect of model choices will help highlight warping as a key phenomenon in neural network computation.
>
> **References**
>
> \[1\] Valente, A., Pillow, J. W., & Ostojic, S. (2022). Extracting computational mechanisms from neural data using low-rank RNNs. Advances in Neural Information Processing Systems, 35, 24072-24086.
>
> \[2\] Mastrogiuseppe, F., & Ostojic, S. (2018). Linking connectivity, dynamics, and computations in low-rank recurrent neural networks. Neuron, 99(3), 609-623.
>
> \[3\] Beiran, M., Meirhaeghe, N., Sohn, H., Jazayeri, M., & Ostojic, S. (2023). Parametric control of flexible timing through low-dimensional neural manifolds. Neuron, 111(5), 739-753.
>
> \[4\] Jazayeri, M., & Ostojic, S. (2021). Interpreting neural computations by examining intrinsic and embedding dimensionality of neural activity. Current opinion in neurobiology, 70, 113-120.

---

> > ### Comment · Reviewer_aFgg · 2025-08-05
> >
> > We thank the authors for their extensive response to our review.
> >
> > The description of the relationship to low rank RNNs is exceptionally well written and very much appreciated. We agree with the authors that this would be a benefit to include in the paper, and would be beneficial to the computational neuroscience community at large.
> >
> > We additionally appreciate the authors' additional experiments on activation functions. Including this in the appendix would also help readers understand the proposed model better.
> >
> > In consideration of this response, we will maintain our score, but lean towards a strong accept, and will therefore advocate this way in future discussions. We thank the authors again for their time.

---

> > > ### Author Response · Authors · 2025-08-05
> > >
> > > We deeply thank the reviewer for their very positive feedback and for their support in future discussions.

---

### Official Review · Reviewer_RgJi · 2025-07-01

**Clarity:** 3
**Significance:** 3
**Originality:** 2
**Rating:** 4
**Confidence:** 4

**Summary:**

In this paper authors use the framework of Riemannian geometry to analyse how RNNs compute through their dynamics. Through their analyses they identify that the computations of RNNs can be captured through the warping of their representational spaces. Authors trace this warping across a couple of established examples of RNN computations.

**Questions:**

Framework definition: In the abstract authors write "To test this hypothesis, we develop a Riemannian geometric framework" -- please comment on what you mean with developing a framework, given that Riemannian geometry has been used already. What specific organizing principles, predictive capabilities, or systematic methodologies does your framework provide beyond applying existing mathematical tools? This links to the criteria of clarity and originality.

Metric definition and consistency: Is your view of your work that you specifically introduce a metric for warping that can be applied across examples? As stated in "We introduce the representational metric defined as the pullback of the metric on the neural space to the input manifold". I am struggling to see where you define a metric which is then coherently used across examples. Please clarify the definition of the metric and explain how this metric highlights mechanisms of computations in later examples. This links to the criteria of clarity and quality.

Mechanistic insights: My reading is that your investigations did not directly reveal any new mechanism about computations in dynamics but more focuses on the introduction of a metric? Please correct me if you have a specific theoretical result that I am overlooking. This links to significance.

Overall contributions: Please provide an overall conclusion statement of your learning across investigations at the start of the 'Conclusions' section. Currently you only mention how Riemannian geometry can be applied across use cases, but it is unclear to me how your work contributes to this application of Riemannian geometry beyond demonstrating that existing mathematical tools can be applied to RNNs. This links to clarity and significance.

Additionally, note typo in line 165: “neurosciencce”.

**Ethical Concerns:**

["NO or VERY MINOR ethics concerns only"]

**Final Justification:**

This work attempts to unify established examples of computations of RNNs through a coherent perspective of manifold warping. Its an interesting and well done body of work and we thank the authors for engaging during the discussion phase.

**Limitations:**

Yes.

**Quality:**

3

**Strengths And Weaknesses:**

Strengths

Attempt to unify established examples of computations of RNNs through a coherent perspective of manifold warping

Strong theoretical analyses of the expected dimensionality of RNN signals as a function of input signals

Systematic application of differential geometric tools to RNN analysis

Beautiful presentation of your results.

Weaknesses

Why is this a new framework?: Authors claim to introduce a new framework but fail to explain in detail how their usage of Riemannian geometry goes beyond existing approaches for analysing dynamic representations.

Inconsistent metric definition: The specific metric they develop based on Riemannian geometry claims to be based on pullback construction, but they fail to define or consistently apply a single coherent metric across analyses. Different sections seem to use different mathematical objects all labeled as 'representational metrics' without clear relationships between formulations. This plays into a larger issue that each of the individual sections seem to have substance by themselves but I am struggling to identify the links between them. This is not helped by the fact that authors do not provide a joint summary of findings across sections at the start of the discussion.

What new findings are possible with the new framework: While perhaps the authors outline a new way of thinking about dynamic representations, they seem to not be able to use this analysis framework to derive findings which have not been reported in the literature yet. The claimed 'dynamic warping' phenomena mimics established findings about RNN learning dynamics (e.g., emergence of structured attractors, dimensionality reduction during training) using different mathematical vocabulary, without providing new computational insights. For discussion of toroidal representation there are existing works (https://journals.plos.org/ploscompbiol/article?id=10.1371/journal.pcbi.1011852, https://www.nature.com/articles/s41583-023-00693-x). For discussion of collapse of information over non-relevant dimensions, this is already shown in Sussillo's work and also in https://www.biorxiv.org/content/10.1101/2023.04.24.538054v2.abstract. In addition, Riemannian geometry has been used already for analysing (static) representations in https://proceedings.neurips.cc/paper_files/paper/2022/hash/4b3cc0d1c897ebcf71aca92a4a26ac83-Abstract-Conference.html.

---

> ### Author Rebuttal · Authors · 2025-07-31
>
> **Summary.** We thank the reviewer for their insightful comments. We are particularly pleased that the reviewer found our work to have “strong theoretical analyses” and “systematic applications of differential geometry” that we used to “unify established examples of computations of RNNs through a coherent perspective of manifold warping.”
>
> The reviewer raised several questions regarding:
>
> * The relationship between our framework and i) other applications of Riemannian geometry in deep learning and ii) attractor manifolds in dynamical systems
> * What insights about the computation can be gained by studying the geometry of neural networks
>
> We’ve addressed these questions by:
>
> * Improving our introduction section to better highlight our contributions and how it relates to existing literature
> * Performing a new set of validations where we explicitly manipulate the intrinsic geometry of the RNN representation to highlight that warping can be a necessary and sufficient component of task computation
>
> The reviewer will find our point-by-point response below.
>
> ___
> **Link to existing literature.** We thank the reviewer for the additional pointers to the literature. We confirm that neither the methods developed in our work, nor the results we obtained with them, had been previously reported. In particular, our work goes beyond previous studies by characterising the (time-varying) geometry of the manifold of states reached by the system upon all possible (time-varying) task inputs.
>
> We have now cited in our paper the works highlighted by the reviewer and discussed their relationship to our own. We discuss this point-by-point below.
>
> **Link attractor manifolds.** The review mentioned by the reviewer (Langdon et al., 2024, *Nat. Rev. Neurosci.*) primarily discusses fixed point attractor manifolds, and the article on toroidal representation (Matthijs et al., 2024, *PLoS CB*) discusses limit cycles. Because they analyse steady-states and autonomous dynamics, these studies cannot directly account for time-varying inputs as our work can.
>
> 🠢 More generally, methods for studying attractor manifolds can only infer a static manifold geometry, and therefore could not be used to tackle the main claim of our paper that representations dynamically change as task computations unfold.
>
> Sussillo’s work (Mante et al. 2013, *Nature*) characterises variability, at steady-state, in the linearised dynamical system, near a line attractor encoding the context-relevant input. They find that the state of the system in response to pulse inputs decays for context-irrelevant but not relevant inputs.
>
> 🠢 Instead, we i) analyse the full nonlinear system ii) characterise the geometry through the entire duration of the task (including the initial transient far from the line attractor), showing that the representation dynamically warps and, iii) characterise the intrinsic geometry of the manifold, showing both warping of the irrelevant input encoding and warping of the relevant input around its decision boundary.
>
> **Link to collapse.** Michał et al. (2024, *biorXiv*) discusses how deep neural networks with static inputs can collapse their state-space. Instead, we describe how *dynamical systems* receiving *time-varying inputs* can dynamically warp the *intrinsic geometry* of their manifolds.
>
> Yaras et al., (2022, *NeurIPS*) studies the intrinsic geometry of neural representation of spheres in deep neural networks. They attempt to better understand how certain classes can be ignored during classification tasks via the so-called neural collapse phenomenon.
>
> 🠢 Instead, we discuss how dynamical warping of the representation of time-varying inputs can be used by a dynamical system to perform computations.
>
> *Thus, although the works mentioned by the reviewer provide useful context to our own — in that they are part of the literature on using geometry to study neural networks — they do not significantly overlap with ours, neither in the mathematical tools they use, nor in the models they study.*
>
> ___
> **Broader novelty of the framework.** Previous work has suggested that the pullback of the metric from the hidden layer representation of a neural network to the input manifold can provide insights into the computations it performs \[1\]. Such geometric approaches towards understanding neural computations have seen success in several applications ranging from convolutional networks in vision tasks \[2\] to transformers in language tasks \[3\].
>
> Our framework extends this view in that we develop a framework to study the geometry and topology of *dynamical systems*, described as differential equations, and which receive time-varying inputs on a manifold.
>
> * We prove a link between the topology of a manifold of time-varying inputs and the topology of the manifold over which the state of the system lies (Theorems 3.1, S7.3 and Corollary S7.5).
> * We exactly derive the geometry of this manifold, defined as the pullback of the metric from the state-space of the dynamical system to the input manifold as a system of adjoint differential equations (Theorems 3.3, 3.4; Section S7.3).
>
> *To better address the link between our work and existing literature, we have now expanded on the related works section to better highlight how our work relates to other uses of Riemannian geometry in machine learning and computational neuroscience.*
>
> ___
> **Metric definition and consistency.** We confirm to the reviewer that we are using the same metric throughout the work, with the minor caveat that in Fig. 6e we consider the metric of the slice of the manifold at a particular time point (a 2x2 submatrix of the 3x3 matrix of Fig. 6d), and in the rest of the work we consider the full space-time metric of the activity manifold. A general overview of the definition of the metric is presented in section 2, and more specific insights into how this metric is computed for deep neural networks and dynamical systems’ can be found in sections S7.1 and S7.2 of supplementary material.
>
> We stress that this metric defines *the* intrinsic geometry of the neural manifold as it stands in the dynamical system’s state-space, and is not an object we’ve arbitrarily chosen. Other metrics can be placed on the manifold, but they would not match the geometry of the neural manifold within the state-space.
>
> *To better highlight this fact, we have now i) clarified the definition of the metric in the background section, and ii) moved some of the Riemannian geometry background from supplementary material to the main manuscript.*
>
> ___
> **Mechanistic insights.** We acknowledge the reviewer’s comment that our analyses were purely correlative, in that we observe geometries that are linked to particular task variables or computations. For example, in our contextual decision-making RNN, we observe that i) the representation of the context-irrelevant input is compressed, and ii) near the decision boundary of the relevant input space is stretched, corresponding to the network switching from an output of $-1$ to $1$. Yet, there could be other mechanisms to solve the task that do not require these warping phenomena, or that the warping phenomena in themselves do not fully capture the computation performed by the network.
>
> To further highlight that **warping is not an epiphenomenon** but a fundamental mechanism by which networks perform the tasks, we’ve now performed the following analyses in the contextual decision-making RNN:
>
> * To assess whether wrapping was necessary, we’ve now trained a network under the constraint that the diagonal elements of the metric remain of equal magnitude — meaning that the representation cannot compress an irrelevant input.
> * To assess whether warping is sufficient, we’ve now trained a network simply to warp the irrelevant input dimension, before testing the model on the full task.
>
> We’ve done this analysis by constraining the ratio of the diagonal elements of the metric to be either 1 or their value in the trained RNN. To do that, we’ve added a term to the loss: $\lVert c - G_{u_{ctx}u_{ctx}}/G_{u_{ctx-1}u_{ctx-1}}\rVert$ where $G_{u_{ctx}u_{ctx}}=\lVert \partial_{u_{ctx}} x\rVert^2$ and $u_{ctx}$ is the input of the current context while $u_{ctx-1}$ is the input of the opposite context..
>
> |  | Untrained model | Baseline trained model | No warping | Untrained $W$ | Warping- trained $W$ |
> | :---- | :---- | :---- | :---- | :---- | :---- |
> | Test performance (MSE, mean $\\pm$ std) | $0.968 \\pm 0.708$ | $0.001 \\pm 0.002$ | $0.239 \\pm 0.257$ | $0.234 \\pm 0.259$ | $0.001 \\pm 0.001$ |
>
> Interestingly warping seems necessary to perform the task, in that the model constrained not to warp does not converge to a good task performance. Furthermore, more surprisingly, simply training the model on warping is sufficient to reach a performance nearly as good as the baseline model trained on the actual task. This suggests that **warping is a necessary and sufficient element of the computation** of the solution to this task.
>
> ___
>
> **Discussion and broader impacts.** As requested by the reviewer, we have now provided an overall conclusion statement. We summarise and link together results from different sections. We highlight that we have developed new mathematical tools, which extend previous applications of Riemannian geometry in machine learning to the case of dynamical systems receiving time-varying inputs on a low-dimensional manifold. Furthermore, we explain that, using these tools, we can show that warping seems to be a ubiquitous feature of the computations performed by RNNs during classic neuroscience tasks.
>
> ___
>
> **References**
>
> \[1\] Hauser, M., & Ray, A. (2017). Principles of Riemannian geometry in neural networks. *NeurIPS*.
>
> \[2\] Kaul, P., & Lall, B. (2019). Riemannian curvature of deep neural networks. *IEEE*.
>
> \[3\] Li, K. et al. (2023). Emergent world representations: Exploring a sequence model trained on a synthetic task. *ICLR*.

---

> > ### Comment · Reviewer_RgJi · 2025-08-04
> > **Reply**
> >
> > Dear authors,
> >
> > Thank you for responding to my questions and clarifying what you think your contribution to the literature beyond prior works is. I understand your point of analysing time-varying dynamics beyond fixed point attractors as an abstract goal. Correct me if I am wrong in saying that the way that your approach goes beyond existing approaches is that you can analyse what is usually visualised as flow fields towards attractors with a more precise geometric description. I agree that this sounds like an approach that may contribute further to our understanding of neural computations. What I am still unclear about is what you believe is the finding that goes beyond the results of prior investigations, that would then show that your method is indeed picking up on signal otherwise discarded. What you seem to suggest is that your finding around ‘warping’ is the new bit, where you say that the manifold is warped to discard irrelevant inputs (for example line 207). How is this different from what has been reported before in for example Mante 2013 that irrelevant stimuli are ignore by converging into attractor states? I wonder whether part of my confusion is arising because the paper does not directly define the term ‘warping’, if I am not mistaken.

---

> > > ### Author Response · Authors · 2025-08-05
> > >
> > > **“I agree that this sounds like an approach that may contribute further to our understanding of neural computations.”**
> > >
> > > We are grateful the reviewer sees this approach as promising, as furthering our understanding of neural computation is our primary goal.
> > >
> > > ___
> > >
> > > **“\[This approach\] analyses what is usually visualised as flow fields towards attractors with a more precise geometric description”**
> > >
> > > This is correct: instead of visualising flow fields, we study the precise geometry of the manifold of states of the system. However, we stress that our approach allows studying systems that perform computations through their transient dynamics, including systems which do not have any attractors at all.
> > >
> > > Our work also goes beyond visualisation of flow fields since:
> > >
> > > 1. As noted by the reviewer, it precisely quantifies the exact nonlinear geometry of the neural representation, via the metric tensor.
> > > 2. In general, flow fields may be too high dimensional to visualize in a 2 or 3D space, whereas the metric of intrinsically low-dimensional manifolds can be computed for dynamics embedded in arbitrarily high-dimensional spaces.
> > > 3. Many dynamical systems cannot be described in terms of fixed flow fields. For example, the RNN equation $dx/dt \= \-x \+ \\phi(Wx+u(t))$ has a flow field that varies nonlinearly as $u(t)$ is varied. This makes visualising their low-dimensional dynamics in terms of flow fields challenging. Instead, our approach directly captures the time-varying low-dimensional geometry of these systems.
> > >
> > > We’ve now better highlighted these features in the contribution and related works sections.
> > >
> > > ___
> > >
> > > **“How is this different from what has been reported before in for example Mante 2013 that irrelevant stimuli are ignored by converging into attractor states?”**
> > >
> > > Our results depart from Mante et al. 2013 in two key ways:
> > >
> > > 1. The results from Mante et al. require a linearisation around a fixed point and therefore describe the dynamics near steady state. Instead, our results suggest that the warping used to discard irrelevant input variability already starts at early time points, before the system has converged towards any fixed point.
> > > 2. In addition to the compression of the irrelevant stimulus, we also found a previously unreported effect: that space was stretched around the decision boundary for the relevant stimulus. This effect is important to the computation, as it allows small changes in inputs around the decision boundary to drive large changes in RNN state, which in turn correspond to changes from an output of \-1 to 1\.
> > >
> > > We fully acknowledge that there is likely be a connection between linearisation around fixed points (e.g. Mante et al.) and our warping results. Such a connection would relate the geometry of manifolds of states of the system to attractor manifolds, which we believe are both central to neural computations. However, deriving a precise mathematical link between fixed point linearisation and nonlinear warping is non-trivial and beyond the scope of our current submission. We will better address this connection as potential for future work in the Discussion.
> > >
> > > ___
> > >
> > > **“the paper does not directly define the term ‘warping’ ”**
> > >
> > > We thank the reviewer for pointing out that our description of warping may be unclear. We believe it is important for the reader to have both a clear definition and intuition of warping, since it is central to our paper. We define warping as a quickly decaying eigenvalue spectrum of the metric. This means that certain directions of the manifold are compressed and stretched. We saw that it can have different effects such as: i) compressing the representation of irrelevant stimuli and ii) stretching space around the decision boundary for the relevant stimulus.
> > >
> > > To address the reviewer’s comment, we have now moved this definition to the background section, and cited classic differential geometry work discussing warping.

---

> > > > ### Comment · Reviewer_RgJi · 2025-08-08
> > > > **reply**
> > > >
> > > > Dear authors,
> > > >
> > > > thank you for the extensive and helpful discussion during the rebuttal period. Based on these clarifications I will raise my score, though I do want to also highlight that even though the topic of study is quite close to my own expertise that I did not get many of these details from the original manuscript, so I hope that these discussion will help with clarifying details for the camera-ready version. Will increase the score.

---

> > > > > ### Author Response · Authors · 2025-08-08
> > > > >
> > > > > We thank the reviewer for the thorough and constructive discussion, which we believe will make the paper clearer and more accessible to a broader audience.

---

### Official Review · Reviewer_3Hky · 2025-07-03

**Clarity:** 4
**Significance:** 3
**Originality:** 3
**Rating:** 5
**Confidence:** 3

**Summary:**

This work studies the geometry of the neural manifold and how it is shaped by a low-dimensional input manifold in an RNN dynamically. Specifically, the authors define a metric on the manifold that quantifies its shape and the evolution of its geometry across time. There are two main conclusions: contextual inputs warp (stretch) this manifold over time to disregard irrelevant inputs. Second, in the working memory task, the torus manifold is warped across time in the encoding phase, but it remains the same during delay.

**Questions:**

It would be nice to have some more intuition about what the metric represents.

In Fig.4f, representing the warping of the manifold, is it an illustration or an actual calculation of the geodesic gridlines?

Working memory is not just about maintenance but also about the manipulation of neural activity. How would you think the manifold would evolve, and how would your conclusions change if you considered a task with manipulation?

**Ethical Concerns:**

["NO or VERY MINOR ethics concerns only"]

**Final Justification:**

The authors have provided a thorough and convincing rebuttal that addresses all the weaknesses I raised in my initial review. I recommend the paper for acceptance.

**Limitations:**

yes

**Quality:**

3

**Strengths And Weaknesses:**

**Strengths**:

The paper is very well written and is backed by mathematical proofs for the arguments made. The problem considered is also an important one: studying how neural manifolds evolve over time. The visuals in the figures make the point go across better!


**Weaknesses**:

The main theorem about the outputs of RNN being low-dimensional if the input is low-dimensional seems trivial. Can the authors state if there are any special dynamical systems where this will not be satisfied, and/or explain why this is non-trivial?

The tasks considered in this work are low-dimensional, which might not be true for RNNs designed for ML. I understand the neuroscience motivation behind considering these tasks, but it would be interesting to see if this warping occurs in some task where the manifold is higher dimensional. It might not be feasible here, but a discussion about whether the conclusions drawn here would hold for a higher-dimensional RNN doing something more complicated, or state what the limitations of this approach are.

Although there’s a calculation about the curvature of the torus but it seems no insights were drawn from it. So I’m not entirely sure of what the utility of that calculation is.

---

> ### Author Rebuttal · Authors · 2025-07-31
>
> **Summary.** We thank the reviewer for the helpful and supportive comments. We were particularly pleased that the reviewer found the work to be “very well written” and “backed by mathematical proofs” while “the problem considered is an important one”.
>
> The main comments raised by the reviewer were:
>
> * Questions regarding the applicability of the framework to higher-dimensional tasks and more complex models
> * A request to provide better intuition regarding the meaning of the metric and manifold dimensionality
> * The effect of manipulations of working memories on warping
>
> We have addressed these points by:
>
> * Adding an example application where we find warping in an SSM trained on a BCI task
> * i) improving the writing of the theoretical results section, ii) revising the schematic of figure 3, iii) moving the supplementary materials results regarding manifold topology to the main manuscript
> * Adding an example of a common neuroscience task which requires computing the difference of numbers stored in working memory
>
> The reviewer will find below our point-by-point responses.
>
> ___
> **Results regarding the dimensionality of the RNN manifold.** Theorem 3.1 introduces a bound on the dimensionality of the manifold of states a dynamical system can take, as a function of the dimensionality of its inputs. We emphasise that this result is novel and non-trivial because 1\) we consider the intrinsic dimensionality of the entire manifold of reachable states of the system, including transients far from steady state 2\) we relate this dimensionality to the dimensionality of the input manifold in the space of functions, rather than input state space.
>
> Moreover, this theorem is in fact a corollary of a stronger theorem we derived in supplementary material (Theorem S7.3 and Corollary S7.5) that relates the topology of the manifold of inputs to the topology of the manifold of states of the dynamical system. In particular, these are related by a projection operator, which places strong constraints on the possible topologies the dynamical system state can take.
>
> *We thank the reviewer for pointing out that, as presented, the dimensionality theorem may seem trivial; we have therefore i) moved the topology theorem back in the main manuscript, and ii) mentioned the dimensionality result as a corollary.*
>
> **Special cases where this fails.** A key insight here is that the manifold of states to which the dynamical system is constrained is determined by the dimensionality of the manifold of time-varying input functions to the system. This contrasts with previous work which had focused on the dimensionality of the inputs *in state space at particular time points*, where this relationship to the dynamical systems’ dimensionality generally doesn’t hold. For example, there exists dynamical systems for which a 1D input can stir a system into any possible state (so-called “controllable systems”).
>
> This also highlights a way in which the theorem can fail: if the assumption that we evaluate the system for a finite time is violated. In general, one can design a dynamical system and a single input function such that in the limit of large times the trajectory of the system is a space-filling curve. This means that although for finite time a dynamical system receiving a single input has solution a 1D manifold — its trajectory over time — in infinite time its trajectory can fill a dense region of space.
>
> ___
>
> **Applicability of the framework to more complex tasks.** We thank the reviewer for asking whether the framework can be applied to tasks with higher-dimensional inputs and systems. Our framework is in principle applicable to higher-dimensional tasks and other dynamical systems typically used in ML (e.g. SSMs, NeuralODEs). The key is to define an insightful manifold of input functions to study.
>
> Previous work has used the pullback of the metric to characterise the geometry of hidden layer activation of deep networks \[1\], for example networks trained on large image datasets (e.g. AlexNet on ImageNet) \[2\]. Despite the set of images living on a relatively high-dimensional manifold, these studies focused on the representation of low-dimensional manifold of possible orientations and positions of a fixed object in the image — which is a slice of the full “manifold of all images”. Similar approaches have since been applied to LLMs \[3\] and other architecture/tasks.
>
> *Our key theoretical contribution is to provide a non-trivial generalisation of this Riemannian geometric view on neural representations to dynamical systems. In general, one could take a dynamical system trained on large language, audio or video time series data, and study the geometry of the representation of a particular manifold of time-varying inputs to the system.*
>
> **New SSM model on a BCI task.** To address the important question raised by the reviewer regarding applicability to more complex tasks, we added an additional model of a state-space model (SSM) trained on a brain-computer interface (BCI) task. We chose this BCI task to remain within the neuroscience theme of the paper while having a system with higher-dimensional input data time series (brain recordings).
>
> We based our model on a simplified version of a recent SSM architecture for neural data \[4\]. The model receives 256-dimensional human neural data time series which it must map to the velocity of a cursor on a screen. Because it is central to BCI control, we chose to study how the manifold of possible instantaneous velocities are encoded by the SSM during the task. In particular, in part of this dataset, the human subject is instructed to draw lines in different directions at different speeds from the center of the screen. This is a $S^1 \\times R$ manifold representing the possible orientations and speeds of the cursor.
>
> *Using our framework on this SSM trained on a BCI task, we found that neural activity warps along the time dimension across different velocities. In other words, to produce faster cursor movements neural activity speeds up along the trajectory in the neural state space.*
>
> ___
> **Insights gained from the curvature of the torus.** We thank the reviewer for bringing to our attention the fact that the current writing doesn’t adequately motivate the analysis of the curvature of the torus. The motivation for this analysis was the long line of work suggesting that working memories are encoded in orthogonal spaces of neural activity space, which implies a flat manifold (zero curvature). Our analysis suggests that, at least in this classic RNN model of working memory, this view is inadequate: the torus is curved, and dynamically warps across the task, requiring a time-resolved analysis of transient states to be fully characterised, as per our method.
>
> *We have now better motivated the analysis of the curvature of the torus by more thoroughly discussing the existing literature on orthogonal representations of multiple items in working memory.*
>
> ___
> **Intuition about the meaning of the metric.** The Riemannian metric characterises the intrinsic geometry of the manifold over which the state of the RNN lies. The representational “pullback” metric is defined in terms of the dot products of the partial derivatives $\\partial\_{u\_i}  \\mathbf{x}$ and $\\partial\_{t} \\mathbf{x}$ which describe how the state of the system changes when the input function or time changes within the manifold of input functions.
>
> The off-diagonal term can be interpreted as representing how correlated the encoding of the inputs are: if $G\_{u_1,u_2}=$$\\partial\_{u_1}  \\mathbf{x}$ $\\cdot$ $\\partial\_{u_2}  \\mathbf{x} \= 0 $ this means that the two variables are have an orthogonal encoding.
>
> The diagonal terms can be be interpreted as the degree of space warping: in the extreme case where $G\_{u_1,u_1}=||\\partial\_{u_1} \mathbf{x}||^2=0$ (meaning that the state of the dynamical system doesn’t change with this input), the state manifold is warped (compressed) such that the corresponding dimension of the manifold disappears (has zero volume).
>
> *To give the reader better intuition about the meaning of the metric, we have now included this more intuitive discussion, and revised the schematic from figure 3\.*
>
> ___
> **Is Fig.4f the actual warping?** We confirm to the reviewer that it is indeed the actual calculation of the geodesics. We were particularly pleased that the result so clearly highlighted that space was stretched around the decision boundary near the decision time.
>
> ___
> **Effect of manipulation of working memories.** This is a great point raised by the reviewer. In our task, all the network has to do is to store in memory and then retrieve the inputs it is presented. Yet, there are several other neuroscience tasks — and real-world scenarios — which involve the manipulation of those memories (to perform calculations or else).
>
> *To better highlight the effect of manipulations of working memory we have now applied our framework to an RNN trained on a new task, where a network must perform a subtraction over numbers stored in memory \[5\]. We found that the manifold starts two dimensional (to represent both numbers in memory), and then dynamically warps and collapses to have a nearly rank-1 metric (i.e. being effectively 1-dimensional), to encode the difference of these numbers.*
>
> ___
>
> **References**
>
> \[1\] Hauser, M., & Ray, A. (2017). Principles of Riemannian geometry in neural networks. *NeurIPS*.
>
> \[2\] Kaul, P., & Lall, B. (2019). Riemannian curvature of deep neural networks. *IEEE*.
>
> \[3\] Li, K. et al. (2023). Emergent world representations: Exploring a sequence model trained on a synthetic task. *ICLR*.
>
> \[4\] Ryoo, A. H. W. et al. (2025). Generalizable, real-time neural decoding with hybrid state-space models. *NeurIPS*.
>
> \[5\] Schuessler, F. et al. (2020). The interplay between randomness and structure during learning in RNNs.  *NeurIPS*.

---

> > ### Comment · Reviewer_3Hky · 2025-08-03
> >
> > I thank the authors for their detailed rebuttal and for providing helpful clarifications on the theoretical results and the metric's interpretation. The addition of the BCI and working memory manipulation tasks directly addresses my main concerns and will strengthen the paper's contributions.

---

> > > ### Author Response · Authors · 2025-08-05
> > >
> > > Thank you for your thoughtful feedback and for acknowledging our clarifications. We're glad the additions addressed your concerns.

---

### Note · Authors · 2025-08-16

Thanks to the reviewers for the thoughtful discussion and for their positive feedback regarding the novelty and interest of our submission. We are particularly pleased that, following the additional simulations and clarifications provided in our rebuttal, the reviewers unanimously acknowledged that we had addressed their comments. We are especially grateful for the suggested analyses and literature links, which have further highlighted the significance of our work, for both the machine learning and computational neuroscience communities. We thus hope our framework will provide a theoretical foundation for using Riemannian geometry to study how nonlinear dynamics warp neural representations.

---

### Decision · Program_Chairs · 2025-09-17

**Decision:**

Accept (poster)

**Comment:**

This paper proposes a framework based on Riemann geometry to argue that RNNs perform computations by dynamically warping the manifolds of their internal representations. Reviewers found the work to be well-presented and mathematically sound but initially raised concerns about its novelty and the simplicity of the tasks. In rebuttal, the authors successfully addressed most technical critiques by clarifying their contributions and adding new experiments on a more complex BCI task and causal analyses of the warping mechanism. While these changes improved the completeness of the analysis, the continued narrow focus on neuroscience-themed problems may limit its impact on a broader audience. Nevertheless, the community may find the techniques and perspective interesting, and I recommend the paper for acceptance.